# No Fear of Heterogeneity: Classifier Calibration for Federated Learning with Non-IID Data

**Mi Luo**[1], **Fei Chen**[2], **Dapeng Hu**[1], **Yifan Zhang**[1], **Jian Liang**[*3], **Jiashi Feng**[*1]

[1]National University of Singapore    [2]Huawei Noah's Ark Lab
[3]Institute of Automation, Chinese Academy of Sciences (CAS)

{romyluo7, liangjian92, jshfeng}@gmail.com
chen.f@huawei.com, {dapeng.hu, yifan.zhang}@u.nus.edu

## Abstract

A central challenge in training classification models in the real-world federated system is learning with non-IID data. To cope with this, most of the existing works involve enforcing regularization in local optimization or improving the model aggregation scheme at the server. Other works also share public datasets or synthesized samples to supplement the training of under-represented classes or introduce a certain level of personalization. Though effective, they lack a deep understanding of how the data heterogeneity affects each layer of a deep classification model. In this paper, we bridge this gap by performing an experimental analysis of the representations learned by different layers. Our observations are surprising: (1) there exists a greater bias in the classifier than other layers, and (2) the classification performance can be significantly improved by post-calibrating the classifier after federated training. Motivated by the above findings, we propose a novel and simple algorithm called *Classifier Calibration with Virtual Representations* (CCVR), which adjusts the classifier using virtual representations sampled from an approximated gaussian mixture model. Experimental results demonstrate that CCVR achieves state-of-the-art performance on popular federated learning benchmarks including CIFAR-10, CIFAR-100, and CINIC-10. We hope that our simple yet effective method can shed some light on the future research of federated learning with non-IID data.

## 1   Introduction

The rapid advances in deep learning have benefited a lot from large datasets like [1]. However, in the real world, data may be distributed on numerous mobile devices and the Internet of Things (IoT), requiring decentralized training of deep networks. Driven by such realistic needs, federated learning [2, 3, 4] has become an emerging research topic where the model training is pushed to a large number of edge clients and the raw data never leave local devices.

A notorious trap in federated learning is training with non-IID data. Due to diverse user behaviors, large heterogeneity may be present in different clients' local data, which has been found to result in unstable and slow convergence [5] and cause suboptimal or even detrimental model performance [6, 7]. There have been a plethora of works exploring promising solutions to federated learning on non-IID data. They can be roughly divided into four categories: 1) client drift mitigation [5, 8, 9, 10], which modifies the local objectives of the clients, so that the local model is consistent with the global model to a certain degree; 2) aggregation scheme [11, 12, 13, 14, 15], which improves the model fusion mechanism at the server; 3) data sharing [6, 16, 17, 18], which introduces public datasets or synthesized data to help construct a more balanced data distribution on the client or on the server;

---

*corresponding author.

4) personalized federated learning [19, 20, 21, 22], which aims to train personalized models for individual clients rather than a shared global model.

However, as suggested by [7], existing algorithms are still unable to achieve good performance on image datasets with deep learning models, and could be no better than vanilla FedAvg [2]. To identify the reasons behind this, we perform a thorough experimental investigation on each layer of a deep neural network. Specifically, we measure the Centered Kernel Alignment (CKA) [23] similarity between the representations from the same layer of different clients' local models. The observation is thought-provoking: comparing different layers learned on different clients, the classifier has the lowest feature[2] similarity across different local models.

Motivated by the above discovery, we dig deeper to study the variation of the weight of the classifier in federated optimization, and confirm that the classifier tends to be biased to certain classes. After identifying this devil, we conduct several empirical trials to debias the classifier via regularizing the classifier during training or calibrating classifier weights after training. We surprisingly find that post-calibration strategy is particularly useful — with only a small fraction of IID data, the classification accuracy is significantly improved. However, this approach cannot be directly deployed in practice since it infringes the privacy rule in federated learning.

Based on the above findings and considerations, we propose a novel and privacy-preserving approach called Classifier Calibration with Virtual Representations (CCVR) which rectifies the decision boundaries (the classifier) of the deep network after federated training. CCVR generates virtual representations based on an approximated Gaussian Mixture Model (GMM) in the feature space with the learned feature extractor. Experimental results show that CCVR achieves significant accuracy improvements over several popular federated learning algorithms, setting the new state-of-the-art on common federated learning benchmarks like CIFAR-10, CIFAR-100 and CINIC-10.

To summarize, our contributions are threefold: (1) We present the first systematic study on the hidden representations of different layers of neural networks (NN) trained with FedAvg on non-IID data and provide a new perspective of understanding federated learning with heterogeneous data. (2) Our study reveals an intriguing fact that the primary reason for the performance degradation of NN trained on non-IID data is the classifier. (3) We propose CCVR (Classifier Calibration with Virtual Representations) — a simple and universal classifier calibration algorithm for federated learning. CCVR is built on top of the off-the-shelf feature extractor and requires no transmission of the representations of the original data, thus raising no additional privacy concern. Our empirical results show that CCVR brings considerable accuracy gains over vanilla federated learning approaches.

## 2 Related Work

Federated learning [2, 3, 4] is a fast-growing research field and remains many open problems to solve. In this work, we focus on addressing the non-IID quagmire [6, 24]. Relevant works have pursued the following four directions.

**Client Drift Mitigation.** FedAvg [2] has been the *de facto* optimization method in the federated setting. However, when it is applied to the heterogeneous setting, one key issue arises: when the global model is optimized with different local objectives with local optimums far away from each other, the average of the resultant client updates (the server update) would move away from the true global optimum [9]. The cause of this inconsistency is called 'client drift'. To alleviate it, FedAvg is compelled to use a small learning rate which may damage convergence, or reduce the number of local iterations which induces significant communication cost [25]. There have been a number of works trying to mitigate 'client drift' of FedAvg from various perspectives. FedProx [5] proposes to add a proximal term to the local objective which regularizes the euclidean distance between the local model and the global model. MOON [8] adopts the contrastive loss to maximize the agreement of the representation learned by the local model and that by the global model. SCAFFOLD [9] performs 'client-variance reduction' and corrects the drift in the local updates by introducing control variates. FedDyn [10] dynamically changes the local objectives at each communication round to ensure that the local optimum is asymptotically consistent with the stationary points of the global objective. FedIR [26] applies importance weight to the local objective, which alleviates the imbalance caused by non-identical class distributions among clients.

---

[2]We use the terms representation and feature interchangeably.

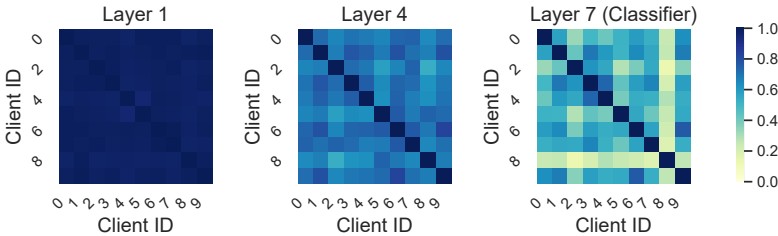

Figure 1: CKA similarities of three different layers of different 'client model-client model' pairs.

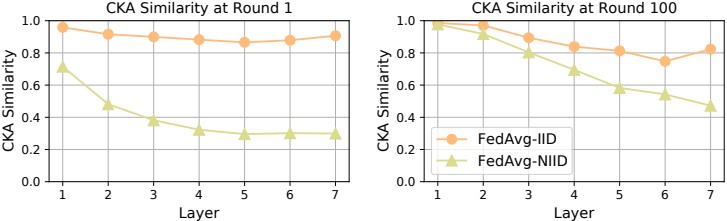

Figure 2: The means of the CKA similarities of different layers in different local models.

**Aggregation Scheme.** A fruitful avenue of explorations involves improvements at the model aggregation stage. These works are motivated by three emerging concerns. First, oscillation may occur when updating the global model using gradients collected from clients with a limited subset of labels. To alleviate it, [11] proposes FedAvgM which adopts momentum update on the server-side. Second, element-wise averaging of weights may have drastic negative effects on the performance of the averaged model. [12] shows that directly averaging local models that are learned from totally distinct data distributions cannot produce a global model that performs well on the global distribution. The authors further propose FedDF that leverages unlabeled data or artificial samples generated by GANs [27] to distill knowledge from the local models. [13] considers the setting where each client performs variable amounts of local works and proposes FedNova which normalizes the local updates before averaging. Third, a handful of works [14, 15] believe that the permutation invariance of neural network parameters may cause neuron mismatching when conducting coordinate-wise averaging of model weights. So they propose to match the parameters of local models while aggregating.

**Data Sharing.** The key motivation behind data sharing is that a client cannot acquire samples from other clients during local training, thus the learned local model under-represents certain patterns or samples from the absent classes. The common practices are to share a public dataset [6], synthesized data [16, 17] or a condensed version of the training samples [18] to supplement training on the clients or on the server. This line of works may violate the privacy rule of federated learning since they all consider sharing raw input data of the model, either real data or artificial data.

**Personalized Federated Learning.** Different from the above directions that aim to learn a single global model, another line of research focuses on learning personalized models. Several works aim to make the global model customized to suit the need of individual users, either by treating each client as a task in meta-learning [19, 28, 20, 29] or multi-task learning [30], or by learning both global parameters for all clients and local private parameters for individual clients [21, 31, 32]. There are also heuristic approaches that divide clients into different clusters based on their learning tasks (objectives) and perform aggregation only within the cluster [33, 34, 22, 35].

In this work, we consider training a single global classification model. To the best of our knowledge, we are the first to decouple the representation and classifier in federated learning — calibrating classifier after feature learning. Strictly speaking, our proposed CCVR algorithm does not fall into any aforementioned research direction but can be readily combined with most of the existing federated learning approaches to achieve better classification performance.

## 3 Heterogeneity in Federated Learning: The Devil Is in Classifier

### 3.1 Problem Setup

We aim to collaboratively train an image classification model in a federated learning system which consists of $K$ clients indexed by $[K]$ and a central server. Client $k$ has a local dataset $\mathcal{D}^k$, and we set

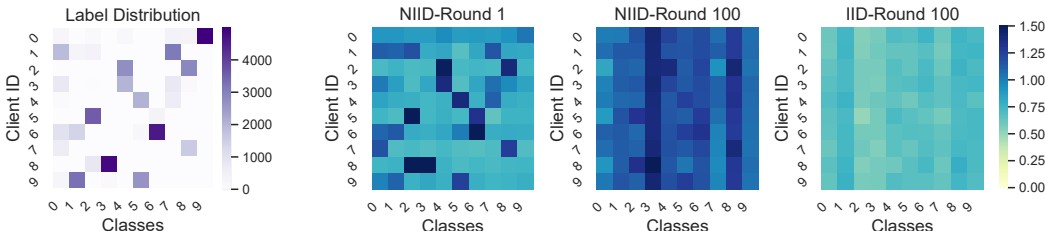

Figure 3: Label distribution of CIFAR-10 across clients (the first graph) and the classifier weight norm distribution across clients in different rounds and data partitions (the three graphs on the right).

$\mathcal{D} = \bigcup_{k \in [K]} \mathcal{D}^k$ as the whole dataset. Suppose there are $C$ classes in $\mathcal{D}$ indexed by $[C]$. Denote by $(\boldsymbol{x}, y) \in \mathcal{X} \times [C]$ a sample in $\mathcal{D}$, where $\boldsymbol{x}$ is an image in the input space $\mathcal{X}$ and $y$ is its corresponding label. Let $\mathcal{D}_c^k = \{(\boldsymbol{x}, y) \in \mathcal{D}^k : y = c\}$ be the set of samples with ground-truth label $c$ on client $k$. We decompose the classification model into a deep feature extractor and a linear classifier. Given a sample $(\boldsymbol{x}, y)$, the feature extractor $f_{\boldsymbol{\theta}} : \mathcal{X} \to \mathcal{Z}$, parameterized by $\boldsymbol{\theta}$, maps the input image $\boldsymbol{x}$ into a feature vector $\boldsymbol{z} = f_{\boldsymbol{\theta}}(\boldsymbol{x}) \in \mathbb{R}^d$ in the feature space $\mathcal{Z}$. Then the classifier $g_{\boldsymbol{\varphi}} : \mathcal{Z} \to \mathbb{R}^C$, parameterized by $\boldsymbol{\varphi}$, produces a probability distribution $g_{\boldsymbol{\varphi}}(\boldsymbol{z})$ as the prediction for $\boldsymbol{x}$. Denote by $\boldsymbol{w} = (\boldsymbol{\theta}, \boldsymbol{\varphi})$ the parameter of the classification model.

Federated learning proceeds through the communication between clients and the server in a round-by-round manner. In round $t$ of the process, the server sends the current model parameter $\boldsymbol{w}^{(t-1)}$ to a set $U^{(t)}$ of selected clients. Then each client $k \in U^{(t)}$ locally updates the received parameter $\boldsymbol{w}^{(t-1)}$ to $\boldsymbol{w}_k^{(t)}$ with the following objective:

$$\min_{\boldsymbol{w}_k^{(t)}} \mathbb{E}_{(\boldsymbol{x}, y) \sim \mathcal{D}^k} [\mathcal{L}(\boldsymbol{w}_k^{(t)}; \boldsymbol{w}^{(t-1)}, \boldsymbol{x}, y)], \tag{1}$$

where $\mathcal{L}$ is the loss function. Note that $\mathcal{L}$ is algorithm-dependent and could rely on the current global model parameter $\boldsymbol{w}^{(t-1)}$ as well. For instance, FedAvg [2] computes $\boldsymbol{w}_k^{(t)}$ by running SGD on $\mathcal{D}^k$ for a number of epochs using the cross-entropy loss, with initialization of the parameter set to $\boldsymbol{w}^{(t-1)}$; FedProx [5] uses the cross entropy loss with an $L_2$-regularization term to constrain the distance between $\boldsymbol{w}_k^{(t)}$ and $\boldsymbol{w}^{(t-1)}$; MOON [8] introduces a contrastive loss term to address the feature drift issue. In the end of round $t$, the selected clients send the optimized parameter back to the server and the server updates the parameter by aggregating heterogeneous parameters as follows,

$$\boldsymbol{w}^{(t)} = \sum_{k \in U^{(t)}} p_k \boldsymbol{w}_k^{(t)}, \text{ where } p_k = \frac{|\mathcal{D}^k|}{\sum_{k' \in U^{(t)}} |\mathcal{D}^{k'}|}.$$

### 3.2 A Closer Look at Classification Model: Classifier Bias

To vividly understand how non-IID data affect the classification model in federated learning, we perform an experimental study on heterogeneous local models. For the sake of simplicity, we choose CIFAR-10 with 10 clients which is a standard federated learning benchmark, and a convolutional neural network with 7 layers used in [8]. As for the non-IID experiments, we partition the data according to the Dirichlet distribution with the concentration parameter $\alpha$ set as $0.1$. More details are covered in the Appendix. To be specific, for each layer in the model, we leverage the recently proposed Centered Kernel Alignment (CKA) [23] to measure the similarity of the output features between two local models, given the same input testing samples. CKA outputs a similarity score between 0 (not similar at all) and 1 (identical). We train the model with FedAvg for 100 communication rounds and each client optimizes for 10 local epochs at each round.

We first selectively show the pairwise CKA features similarity of three different layers across local models in Figure 1. Three compared layers here are the first layer, the middle layer (Layer 4), and the last layer (the classifier), respectively. Interestingly, we find that features outputted by the deeper layer show lower CKA similarity. It indicates that, for federated models trained on non-IID data, the deeper layers have heavier heterogeneity across different clients. By averaging the pairwise CKA

Table 1: Accuracy@1 (%) on CIFAR-10 with different degrees of heterogeneity.

| Method | $\alpha = 0.5$ | $\alpha = 0.1$ | $\alpha = 0.05$ |
|---|---|---|---|
| FedAvg | 68.62±0.77 | 58.55±0.98 | 52.33±0.43 |
| FedAvg + clsnorm | 69.65±0.35 (↑ 1.03) | 58.94±0.08 (↑ 0.39) | 51.74±4.02 (↓ 0.59) |
| FedAvg + clsprox | 68.82±0.75 (↑ 0.20) | 59.04±0.70 (↑ 0.49) | 52.38±0.78 (↑ 0.05) |
| FedAvg + clsnorm + clsprox | 68.75±0.75 (↑ 0.13) | 58.80±0.30 (↑ 0.25) | 52.39±0.24 (↑ 0.06) |
| FedAvg + calibration (whole data) | 72.51±0.53 (↑ 3.89) | 64.70±0.94 (↑ 6.15) | 57.53±1.00 (↑ 5.20) |

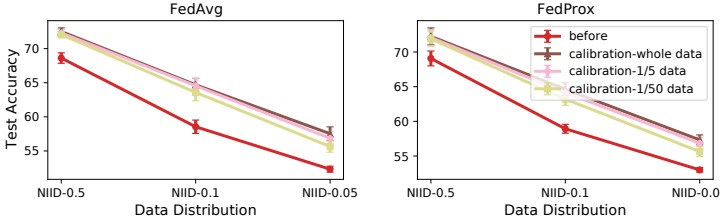

Figure 4: The effect of classifier calibration using different amounts of data.

features similarity in Figure 1, we can obtain a single value to approximately represent the similarity of the feature outputs by each layer across different clients. We illustrate the approximated layer-wise features similarity in Figure 2. The results show that the models trained with non-IID data have consistently lower feature similarity across clients for all layers, compared with those trained on IID data. The primary finding is that, for non-IID training, the classifier shows the lowest features similarities, among all the layers. The low CKA similarities of the classifiers imply that the local classifiers change greatly to fit the local data distribution.

To perform a deeper analysis on the classifier trained on non-IID data, inspired by [36], we illustrate the $L_2$ norm of the local classifier weight vectors in Figure 3. We observe that the classifier weight norms would be biased to the class with more training samples at the initial training stage. At the end of the training, models trained on non-IID data suffer from a much heavier biased classifier than the models trained on IID data.

Based on the above observations about the classifier, we hypothesize that: because the classifier is the closest layer to the local label distribution, it can be easily biased to the heterogeneous local data, reflected by the low features similarity among different local classifiers and the biased weight norms. Furthermore, we believe that debiasing the classifier is promising to directly improve the classification performance.

### 3.3 Classifier Regularization and Calibration

To effectively debias the classifier, we consider the following regularization and calibration methods.

*Classifier Weight L2-normalization.* To eliminate the bias in classifier weight norms, we normalize the classifier weight vectors during the training and the inference stage. We abbreviate it to 'clsnorm'. In particular, the classifier is a linear transformation with weight $\boldsymbol{\varphi} = [\boldsymbol{\varphi}_1, \ldots, \boldsymbol{\varphi}_C]$, followed by normalization and softmax. Given a feature $\boldsymbol{z}$, the output of the classifier is

$$g_{\boldsymbol{\varphi}}(\boldsymbol{z})_i = \frac{e^{\boldsymbol{\varphi}_i^T \boldsymbol{z}/||\boldsymbol{\varphi}_i||}}{\sum_{i'=1}^{C} e^{\boldsymbol{\varphi}_{i'}^T \boldsymbol{z}/||\boldsymbol{\varphi}_{i'}||}}, \quad \forall i \in [C].$$

*Classifier Quadratic Regularization.* Beyond restricting the weight norms of classifier, we also consider adding a proximal term similar to [5] only to restrict the classifier weights to be close to the received global classifier weight vectors from the server. We write it as 'clsprox' for short. The loss function in Eq. (1) can be specified as

$$\mathcal{L}(\boldsymbol{w}_k^{(t)}; \boldsymbol{w}^{(t-1)}, \boldsymbol{x}, y) = \ell(g_{\boldsymbol{\varphi}_k^{(t)}}(f_{\boldsymbol{\theta}_k^{(t)}}(\boldsymbol{x})), y) + \frac{\mu}{2}||\boldsymbol{\varphi}_k^{(t)} - \boldsymbol{\varphi}^{(t-1)}||^2,$$

where $\ell$ is the cross-entropy loss and $\mu$ is the regularization factor.

*Classifier Post-calibration with IID Samples.* In addition to regularizing the classifier during federated training, we also consider a post-processing technique to adjust the learned classifier. After the federated training, we fix the feature extractor and calibrate the classifier by SGD optimization with a cross-entropy loss on IID samples. Note that this calibration strategy requires IID raw features collected from heterogeneous clients. Therefore, it can only serve as an experimental study use but cannot be applied to the real federated learning system.

We conduct experiments to compare the above three methods on CIFAR-10 with three different degrees of data heterogeneity and present the results in Table 1. We observe that regularizing the L2-norm of classifier weight (clsnorm) is effective for light data heterogeneity but would have less help or even lead to damages along with the increase of the heterogeneity. Regularizing the classifier parameters (clsprox) is consistently effective but with especially minor improvements. Surprisingly, we find that calibrating the classifier of the FedAvg model with all training samples brings significant performance improvement for all degrees of data heterogeneity.

To further understand the classifier calibration technique, we additionally perform calibrations with different numbers of data samples and different off-the-shelf federated models trained by FedAvg and FedProx. The results are shown in Figure 4 and we observe that data-based classifier calibration performs consistently well, even with $1/50$ training data samples for calibration use. These significant performance improvements after adjusting the classifier strongly verify our aforementioned hypothesis, i.e., the devil is in the classifier.

## 4 Classifier Calibration with Virtual Representations

Motivated by the above observations, we propose Classifier Calibration with Virtual Representations (CCVR) that runs on the server after federated training the global model. CCVR uses virtual features drawn from an estimated Gaussian Mixture Model (GMM), without accessing any real images. Suppose $f_{\widehat{\boldsymbol{\theta}}}$ and $g_{\widehat{\boldsymbol{\varphi}}}$ are the feature extractor and classifier of the global model, respectively, where $\widehat{\boldsymbol{w}} = (\widehat{\boldsymbol{\theta}}, \widehat{\boldsymbol{\varphi}})$ is the parameter trained by a certain federated learning algorithm, e.g. FedAvg. We shall use $f_{\widehat{\boldsymbol{\theta}}}$ to extract features and estimate the corresponding feature distribution, and re-train $g$ using generated virtual representations.

**Feature Distribution Estimation.** For semantics related tasks such as classification, the features learned by deep neural networks can be approximated with a mixture of Gaussian distribution. Theoretically, any continuous distribution can be approximated by using a finite number of mixture of gaussian distributions [37]. In our CCVR, we assume that features of each class in $\mathcal{D}$ follow a Gaussian distribution. The server estimates this distribu-

---

**Algorithm 1:** Virtual Representation Generation

**Input:** Feature extractor $f_{\widehat{\boldsymbol{\theta}}}$ of the global model, number $M_c$ of virtual features for class $c$

1  # Server executes:
2  Send $f_{\widehat{\boldsymbol{\theta}}}$ to clients.
3  # Clients execute:
4  **foreach** *client $k \in [K]$* **do**
5      **foreach** *class $c \in [C]$* **do**
6          Produce $\boldsymbol{z}_{c,k,j} = f_{\widehat{\boldsymbol{\theta}}}(\boldsymbol{x}_{c,k,j})$ for $j$-th sample in $\mathcal{D}_c^k$ for $j \in [N_{c,k}]$.
7          Compute $\boldsymbol{\mu}_{c,k}$ and $\boldsymbol{\Sigma}_{c,k}$ using Eq. (2).
8      **end**
9      Send $\{(\boldsymbol{\mu}_{c,k}, \boldsymbol{\Sigma}_{c,k}) : c \in [C]\}$ to server.
10  **end**
11  # Server executes:
12  **foreach** *class $c \in [C]$* **do**
13      Compute $\boldsymbol{\mu}_c$ and $\boldsymbol{\Sigma}_c$ using Eq. (3) and (4).
14      Draw a set $G_c$ of $M_c$ features from $\mathcal{N}(\boldsymbol{\mu}_c, \boldsymbol{\Sigma}_c)$ with ground truth label $c$.
15  **end**

**Output:** Set of virtual representations $\bigcup_{c \in [C]} G_c$

---

tion by computing the mean $\boldsymbol{\mu}_c$ and the covariance $\boldsymbol{\Sigma}_c$ for each class $c$ of $\mathcal{D}$ using gathered local statistics from clients, without accessing true data samples or their features. In particular, the server first sends the feature extractor $f_{\widehat{\boldsymbol{\theta}}}$ of the trained global model to clients. Let $N_{c,k} = |\mathcal{D}_c^k|$ be the number of samples of class $c$ on client $k$, and set $N_c = \sum_{k=1}^{K} N_{c,k}$. Client $k$ produces features $\{\boldsymbol{z}_{c,k,1}, \dots, \boldsymbol{z}_{c,k,N_{c,k}}\}$ for class $c$, where $\boldsymbol{z}_{c,k,j} = f_{\widehat{\boldsymbol{\theta}}}(\boldsymbol{x}_{c,k,j})$ is the feature of the $j$-th sample in $\mathcal{D}_c^k$, and computes local mean $\boldsymbol{\mu}_{c,k}$ and covariance $\boldsymbol{\Sigma}_{c,k}$ of $\mathcal{D}_c^k$ as:

$$\boldsymbol{\mu}_{c,k} = \frac{1}{N_{c,k}} \sum_{j=1}^{N_{c,k}} \boldsymbol{z}_{c,k,j}, \quad \boldsymbol{\Sigma}_{c,k} = \frac{1}{N_{c,k}-1} \sum_{j=1}^{N_{c,k}} (\boldsymbol{z}_{c,k,j} - \boldsymbol{\mu}_{c,k}) (\boldsymbol{z}_{c,k,j} - \boldsymbol{\mu}_{c,k})^T, \quad (2)$$

Then client $k$ uploads $\{(\boldsymbol{\mu}_{c,k}, \boldsymbol{\Sigma}_{c,k}) : c \in [C]\}$ to server. For the server to compute the global statistics of $\mathcal{D}$, it is sufficient to represent the global mean $\boldsymbol{\mu}_c$ and covariance $\boldsymbol{\Sigma}_c$ using $\boldsymbol{\mu}_{c,k}$'s and $\boldsymbol{\Sigma}_{c,k}$'s for each class $c$. The global mean can be straightforwardly written as

$$\boldsymbol{\mu}_c = \frac{1}{N_c} \sum_{k=1}^{K} \sum_{j=1}^{N_{c,k}} \boldsymbol{z}_{c,k,j} = \sum_{k=1}^{K} \frac{N_{c,k}}{N_c} \boldsymbol{\mu}_{c,k}. \tag{3}$$

For the covariance, note that by definition we have

$$(N_{c,k} - 1)\,\boldsymbol{\Sigma}_{c,k} = \sum_{j=1}^{N_{c,k}} \boldsymbol{z}_{c,k,j} \boldsymbol{z}_{c,k,j}^T - N_{c,k} \cdot \boldsymbol{\mu}_{c,k} \boldsymbol{\mu}_{c,k}^T$$

whenever $N_{c,k} \geq 1$. Then the global covariance can be written as

$$\begin{aligned}
\boldsymbol{\Sigma}_c &= \frac{1}{N_c - 1} \sum_{k=1}^{K} \sum_{j=1}^{N_{c,k}} \boldsymbol{z}_{c,k,j} \boldsymbol{z}_{c,k,j}^T - \frac{N_c}{N_c - 1} \boldsymbol{\mu}_c \boldsymbol{\mu}_c^T \\
&= \sum_{k=1}^{K} \frac{N_{c,k} - 1}{N_c - 1} \boldsymbol{\Sigma}_{c,k} + \sum_{k=1}^{K} \frac{N_{c,k}}{N_c - 1} \boldsymbol{\mu}_{c,k} \boldsymbol{\mu}_{c,k}^T - \frac{N_c}{N_c - 1} \boldsymbol{\mu}_c \boldsymbol{\mu}_c^T.
\end{aligned} \tag{4}$$

**Virtual Representations Generation.** After obtaining $\boldsymbol{\mu}_c$'s and $\boldsymbol{\Sigma}_c$'s, the server generates a set $G_c$ of virtual features with ground truth label $c$ from the Gaussian distribution $\mathcal{N}(\boldsymbol{\mu}_c, \boldsymbol{\Sigma}_c)$. The number $M_c := |G_c|$ of virtual features for each class $c$ could be determined by the fraction $\frac{N_c}{|\mathcal{D}|}$ to reflect the inter-class distribution. See Algorithm 1.

**Classifier Re-Training.** The last step of our CCVR method is classifier re-training using virtual representations. We take out the classifier $g$ from the global model, initialize its parameter as $\widehat{\boldsymbol{\varphi}}$, and re-train the parameter to $\widetilde{\boldsymbol{\varphi}}$ for the objective

$$\min_{\widetilde{\boldsymbol{\varphi}}} \mathbb{E}_{(\boldsymbol{z},y) \sim \bigcup_{c \in [C]} G_c} [\ell(g_{\widetilde{\boldsymbol{\varphi}}}(\boldsymbol{z}), y)],$$

where $\ell$ is the cross-entropy loss. We then obtain the final classification model $g_{\widetilde{\boldsymbol{\varphi}}} \circ f_{\widehat{\boldsymbol{\theta}}}$ consisting of the pre-trained feature extractor and the calibrated classifier.

**Privacy Protection.** CCVR protects privacy at the basic level because each client only uploads their local Gaussian statistics rather than the raw representations. Note that CCVR is just a post-hoc method, so it can be easily combined with some privacy protection techniques [38] to further secure privacy. In the Appendix, we provide an empirical analysis on the privacy-preserving aspect.

## 5 Experiment

### 5.1 Experiment Setup

**Federated Simulation.** We consider image classification task and adopt three datasets from the popular FedML benchmark [39], i.e., CIFAR-10 [40], CIFAR-100 [40] and CINIC-10 [41]. Note that CINIC-10 is constructed from ImageNet [42] and CIFAR-10, whose samples are very similar but not drawn from identical distributions. Therefore, it naturally introduces distribution shifts which is suited to the heterogeneous nature of federated learning. To simulate federated learning scenario, we randomly split the training set of each dataset into $K$ batches, and assign one training batch to each client. Namely, each client owns its local training set. We hold out the testing set at the server for evaluation of the classification performance of the global model. For hyperparameter tuning, we first take out a 15% subset of training set for validation. After selecting the best hyperparameter, we return the validation set to the training set and retrain the model. We are interested in the NIID partitions of the three datasets, where class proportions and number of data points of each client are unbalanced. Following [14, 15], we sample $p_i \sim Dir_K(\alpha)$ and assign a $p_{i,k}$ proportion of the samples from class $i$ to client $k$. We set $\alpha$ as 0.5 unless otherwise specified. For fair comparison, we apply the same data augmentation techniques for all methods.

Table 2: Accuracy@1 (%) on CIFAR-10 with different degrees of heterogeneity ($\alpha \in \{0.5, 0.1, 0.05\}$), CIFAR-100 and CINIC-10.

| | Method | $\alpha = 0.5$ | $\alpha = 0.1$ | $\alpha = 0.05$ | CIFAR-100 | CINIC-10 |
|---|---|---|---|---|---|---|
| No Calibration | FedAvg | 68.62±0.77 | 58.55±0.98 | 52.33±0.43 | 66.25±0.54 | 60.20±2.04 |
| | FedProx | 69.07±1.07 | 58.93±0.64 | 53.00±0.32 | 66.31±0.39 | 60.52±2.07 |
| | FedAvgM | 69.00±1.68 | 59.22±1.14 | 51.98±0.91 | 66.43±0.23 | 60.46±0.73 |
| | MOON | 70.48±0.36 | 57.36±0.85 | 49.91±0.38 | 67.02±0.31 | 65.67±2.10 |
| CCVR (Ours.) | FedAvg | 71.03±0.40 (↑ 2.41) | **62.68±0.54** (↑ 4.13) | 54.95±0.61 (↑ 2.62) | 66.60±0.63 (↑ 0.35) | 69.99±0.54 (↑ 9.79) |
| | FedProx | 70.99±1.21 (↑ 1.92) | 62.60±0.43 (↑ 3.67) | **55.79±1.07** (↑ 2.79) | 66.61±0.48 (↑ 0.30) | 70.05±0.66 (↑ 9.53) |
| | FedAvgM | **71.49±0.88** (↑ 2.49) | 62.64±1.07 (↑ 3.42) | 54.57±0.58 (↑ 2.59) | 66.71±0.16 (↑ 0.28) | **70.87±0.61** (↑ 10.41) |
| | MOON | 71.29±0.11 (↑ 0.81) | 62.22±0.70 (↑ 4.86) | 55.60±0.63 (↑ 5.69) | **67.17±0.37** (↑ 0.15) | 69.42±0.65 (↑ 3.75) |
| Oracle | FedAvg | 72.51±0.53 (↑ 3.89) | 64.70±0.94 (↑ 6.15) | 57.53±1.00 (↑ 5.20) | 66.84±0.50 (↑ 0.59) | **73.47±0.30** (↑ 13.27) |
| | FedProx | 72.26±1.22 (↑ 3.19) | 64.63±0.93 (↑ 5.70) | 57.33±0.72 (↑ 4.33) | 66.68±0.43 (↑ 0.37) | 73.10±0.57 (↑ 12.58) |
| | FedAvgM | **73.30±0.19** (↑ 4.30) | 64.24±1.32 (↑ 5.02) | 57.11±1.08 (↑ 5.13) | 66.94±0.32 (↑ 0.51) | 72.88±0.37 (↑ 12.42) |
| | MOON | 72.05±0.16 (↑ 1.57) | **64.94±0.58** (↑ 7.58) | **58.14±0.47** (↑ 8.23) | **67.56±0.44** (↑ 0.54) | 73.38±0.23 (↑ 7.71) |

**Baselines and Implementation.** We consider comparing the test accuracies of the representative federated learning algorithms FedAvg [2], FedProx [5], FedAvgM [11, 26] and the state-of-the-art method MOON [8] before and after applying our CCVR. For FedProx and MOON, we carefully tune the coefficient of local regularization term $\mu$ and report their best results. For FedAvgM, the server momentum is set to be 0.1. We use a simple 4-layer CNN network with a 2-layer MLP projection head described in [8] for CIFAR-10. For CIFAR-100 and CINIC-10, we adopt MobileNetV2 [43]. For CCVR, to make the virtual representations more Gaussian-like, we apply ReLU and Tukey's transformation before classifier re-training. For Tukey's transformation, the parameter is set to be 0.5. For each dataset, all methods are evaluated with the same model for fair comparison. The proposed CCVR algorithm only has one important hyperparameter, the number of feature samples $M_c$ to generate. Unless otherwise stated, $M_c$ is set to 100, 500 and 1000 for CIFAR-10, CIFAR-100 and CINIC-10 respectively. All experiments run with PyTorch 1.7.1. More details about the implementation and datasets are summarized in the Appendix.

## 5.2 Can classifier calibration improve performance of federated learning?

In Table 2, we present the test accuracy on all datasets before and after applying our CCVR. We also report the results under an ideal setting where the whole data are available for classifier calibration (Oracle). These results indicate the upper bound of classifier calibration.

**CCVR consistently improves all baseline methods.** First, it can be observed that applying classifier calibration increases accuracies for all baseline methods, even with the accuracy gain up to 10.41% on CINIC-10. This is particularly inspiring because CCVR requires no modification to the original federated training process. One can easily get considerable accuracy profits by simply post-processing the trained global model. Comparing the accuracy gains of different methods after applying CCVR and whole data calibration, we find that the accuracies of FedAvg and MOON get the greatest increase. On CINIC-10, the oracle results of FedAvg even outstrip those of all other baselines, implying that FedAvg focuses more on learning high-quality features but ignores learning a fair classifier. It further confirms the necessity of classifier calibration.

## 5.3 In what situation does CCVR work best?

We observe that though there is improvement on CIFAR-100 by applying CCVR, it seems subtle compared with that of other two datasets. This is not surprising, since the final accuracy achieved by classifier calibration is not only dependent on the degree to which the classifier is debiased, but also closely correlated with the quality of pre-trained representations. In CIFAR-100, each class only has 500 training images, so the classification task itself is very difficult and the model may learn representations with low separability. It is shown that the accuracy obtained with CCVR on CIFAR-100 is very close to the upper bound, indicating that CCVR does a good job of correcting the classifier, even if it is provided with a poor feature extractor.

We also note that CCVR achieves huge improvements on CINIC-10. To further analyze the reason of this success and the characteristics of CCVR, we now show the t-SNE visualization [44] of the features learned by FedAvg on CINIC-10 dataset in Figure 5. From the first and second sub-graphs, we can observe that some classes dominate the classification results, while certain classes are rarely

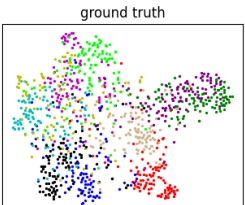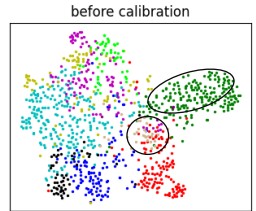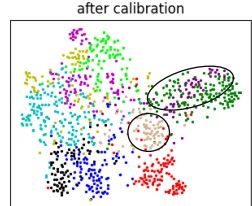

Figure 5: t-SNE visualization of the features learned by FedAvg on CINIC-10. The features are colored by the ground truth and the predictions of the classifier before and after applying CCVR. Best viewed in color.

predicted correctly. For instance, the classifier makes wrong prediction for most of the samples belonging to the grey class. Another evidence showing there exists a great bias in the classifier is that, from the upper right corner of the ground truth sub-graph, we can see that the features colored green and those colored purple can be easily separated. However, due to biases in the classifier, nearly all purple features are wrongly classified as the green class. Observing the third sub-graph, we find that by applying CCVR, these misclassifications are alleviated. We also find that, with CCVR, mistakes are basically made when identifying easily-confused features that are close to the decision boundary rather than a majority of features that belong to certain classes. This suggests that the classifier weight has been adjusted to be more fair to each class. In summary, CCVR may be more effective when applied to the models with good representations but serious classifier biases.

### 5.4 How to forecast the performance of classifier calibration?

We resort to Sliced Wasserstein Distance [45], which is a popular metric to measure the distances between distributions, to quantify the separability of GMM. The experiments are conducted on CIFAR-10 with $\alpha = 0.1$. We first compute the Wasserstein distances between any two mixtures, then we average all the distances to get a mean distance. The farther the distance, the better the separability of GMM. We visualize the relationship between the accuracy gains and the separability of GMM in Figure 6. It is observed that the mean Wasserstein distance of GMM is positively correlated with the accuracy upper bound of classifier calibration. It verifies our claim in Section 5.3: CCVR may be more effective when applied to the models with good (separable) representations. In practice, one can use the mean Wasserstein distance of GMM to evaluate the quality of the simulated representations, as well as to forecast the potential performance of classifier calibration.

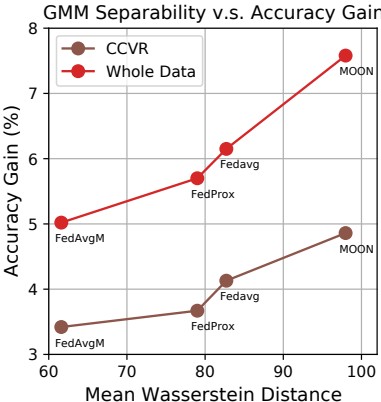

Figure 6: GMM's separability.

### 5.5 How many virtual features to generate?

One important hyperparameter in our CCVR is the number of virtual features $M_c$ for each class $c$ to generate. We study the effect of $M_c$ by tuning it from $\{0, 50, 100, 500, 1000, 2000\}$ on three different partitions of CIFAR-10 ($\alpha \in \{0.05, 0.1, 0.5\}$) when applying CCVR to FedAvg. The results are provided in Figure 7. In general, even sampling only a few features can significantly increase the classification accuracy. Additionally, it is observed that on the two more heterogeneous distributions (the left two sub-graphs), more samples produces higher accuracy. Although results on NIID-0.5 give a similar hint in general, an accuracy decline when using a medium number of virtual samples is observed. This suggests that $M_c$ is more sensitive when faced with a more balanced dataset. This can be explained by the nature of CCVR: utilizing virtual feature distribution to mimic the original feature distribution. As a result, if the number of virtual samples is limited, the simulated distribution may deviates from the true feature distribution. The results on NIID-0.5 implies that this trap could be easier to trigger when CCVR dealing with a more balanced original distribution. To conclude, though CCVR can provide free lunch for federated classification, one should still be very careful when tuning $M_c$ to achieve higher accuracy. Generally speaking, a larger value of $M_c$ is better.

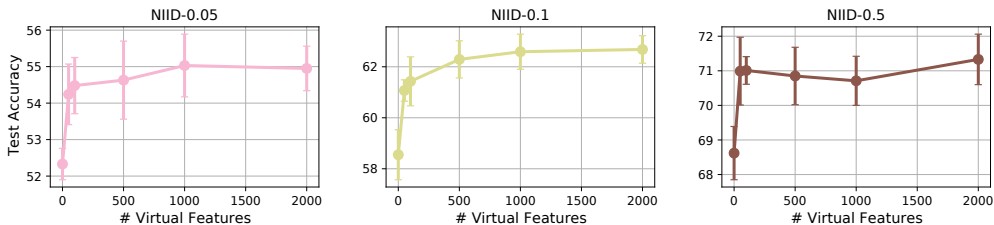

Figure 7: Accuracy@1 (%) of CCVR on CIFAR-10 with different numbers of virtual samples.

### 5.6 Does different levels of heterogeneity affect CCVR's performance?

We study the effect of heterogeneity on CIFAR-10 by generating various non-IID partitions from Dirichlet distribution with different concentration parameters $\alpha$. Note that partition with smaller $\alpha$ is more imbalanced. It can be seen from Table 2 that CCVR steadily improves accuracy for all the methods on all partitions. Typically, the improvements is greater when dealing with more heterogeneous data, implying that the amount of bias existing in the classifier is positively linked with the imbalanceness of training data. Another interesting discovery is that vanilla MOON performs worse than FedAvg and FedProx when $\alpha$ equals to 0.1 or 0.05, but the oracle results after classifier calibration is higher than those of FedAvg and FedProx. It indicates that MOON's regularization on the representation brings severe negative effects on the classifier. As a consequence, MOON learns good representations but poor classifier. In that case, applying CCVR observably improves the original results, making the performance of MOON on par with FedAvg and FedProx.

## 6 Limitations

In this work, we mainly focus on the characteristic of the classifier in federated learning, because it is found to change the most during local training. However, our experimental results show that in a highly heterogeneous setting, only calibrating the classifier still cannot achieve comparable accuracies to that obtained on IID data. This is because the performance of classifier calibration highly relies on the quality of learned representations. Thus, it's more important to learn a good feature space. Our experiments reveal that there may exist a trade-off in the quality of representation and classifier in federated learning on non-IID data. Namely, the methods that gain the greatest benefits from classifier calibration typically learn high-quality representations but poor classifier. We believe this finding is intriguing for future research and there is still a long way to tackling the non-IID quagmire.

Moreover, we mainly focus on the image classification task in this work. Our experiments validate that the Gaussian assumption works well for visual model like CNN. However, this conclusion may not hold for language tasks or for other architectures like LSTM [46] and Transformer [47]. We believe the extensions of this work to other tasks and architectures are worth exploring.

## 7 Conclusion

In this work, we provide a new perspective to understand why the performance of a deep learning-based classification model degrades when trained with non-IID data in federated learning. We first anatomize the neural networks and study the similarity of different layers of the models on different clients through recent representation analysis techniques. We observe that the classifiers of different local models are less similar than any other layer, and there is a significant bias among the classifier. We then propose a novel method called Classifier Calibration with Virtual Representations (CCVR), which samples virtual features from an approximated Gaussian Mixture Model (GMM) for classifier calibration to avoid uploading raw features to the server. Experimental results on three image datasets show that CCVR steadily improves over several popular federated learning algorithms.

## Acknowledgement

We would like to thank the anonymous reviewers for their insightful comments and suggestions.

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
