_{\varphi}(z)_i = \frac{e^{\varphi_i^T z / ||\varphi_i||}}{\sum_{i'=1}^{C} e^{\varphi_{i'}^T z / ||\varphi_{i'}||}}, \quad \forall i \in [C].$$

*Classifier Quadratic Regularization.* Beyond restricting the weight norms of classifier, we also consider adding a proximal term similar to [5] only to restrict the classifier weights to be close to the received global classifier weight vectors from the server. We write it as 'clsprox' for short. The loss function in Eq. (1) can be specified as

$$\mathcal{L}(w_k^{(t)}; w^{(t-1)}, x, y) = \ell(g_{\varphi_k^{(t)}}(f_{\theta_k^{(t)}}(x)), y) + \frac{\mu}{2}||\varphi_k^{(t)} - \

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

# A Derivation of Global Mean and Covariance

**Details for deriving Eq. 3 and 4.** Without loss of generality, we assume $N_{c,k} \geq 1$ and $N_c \geq 2$ for each $c \in [C]$ and $k \in [K]$. The global mean can be straightforwardly written as

$$\boldsymbol{\mu}_c = \frac{1}{N_c} \sum_{k=1}^{K} \sum_{j=1}^{N_{c,k}} \boldsymbol{z}_{c,k,j} = \sum_{k=1}^{K} \frac{N_{c,k}}{N_c} \cdot \frac{1}{N_{c,k}} \sum_{j=1}^{N_{c,k}} \boldsymbol{z}_{c,k,j} = \sum_{k=1}^{K} \frac{N_{c,k}}{N_c} \boldsymbol{\mu}_{c,k}$$

For the covariance, note that for $N_{c,k} \geq 2$ we have

$$\boldsymbol{\Sigma}_{c,k} = \frac{1}{N_{c,k}-1} \sum_{j=1}^{N_{c,k}} \boldsymbol{z}_{c,k,j} \boldsymbol{z}_{c,k,j}^T - \frac{1}{N_{c,k}-1} \sum_{j=1}^{N_{c,k}} \boldsymbol{\mu}_{c,k} \boldsymbol{z}_{c,k,j}^T$$

$$- \frac{1}{N_{c,k}-1} \sum_{j=1}^{N_{c,k}} \boldsymbol{z}_{c,k,j} \boldsymbol{\mu}_{c,k}^T + \frac{1}{N_{c,k}-1} \sum_{j=1}^{N_{c,k}} \boldsymbol{\mu}_{c,k} \boldsymbol{\mu}_{c,k}^T$$

$$= \frac{1}{N_{c,k}-1} \sum_{j=1}^{N_{c,k}} \boldsymbol{z}_{c,k,j} \boldsymbol{z}_{c,k,j}^T - \frac{N_{c,k}}{N_{c,k}-1} \boldsymbol{\mu}_{c,k} \boldsymbol{\mu}_{c,k}^T - \frac{N_{c,k}}{N_{c,k}-1} \boldsymbol{\mu}_{c,k} \boldsymbol{\mu}_{c,k}^T + \frac{N_{c,k}}{N_{c,k}-1} \boldsymbol{\mu}_{c,k} \boldsymbol{\mu}_{c,k}^T$$

$$= \frac{1}{N_{c,k}-1} \sum_{j=1}^{N_{c,k}} \boldsymbol{z}_{c,k,j} \boldsymbol{z}_{c,k,j}^T - \frac{N_{c,k}}{N_{c,k}-1} \boldsymbol{\mu}_{c,k} \boldsymbol{\mu}_{c,k}^T.$$

Rearranging yields

$$(N_{c,k}-1) \boldsymbol{\Sigma}_{c,k} = \sum_{j=1}^{N_{c,k}} \boldsymbol{z}_{c,k,j} \boldsymbol{z}_{c,k,j}^T - N_{c,k} \cdot \boldsymbol{\mu}_{c,k} \boldsymbol{\mu}_{c,k}^T.$$

Note that the above equation holds when $N_{c,k} = 1$ as well, where the mean $\boldsymbol{\mu}_{c,k}$ is equivalent to the single feature $\boldsymbol{z}_{c,k,1}$. Then the global covariance can be written as

$$\boldsymbol{\Sigma}_c = \frac{1}{N_c-1} \sum_{k=1}^{K} \sum_{j=1}^{N_{c,k}} (\boldsymbol{z}_{c,k,j} - \boldsymbol{\mu}_c)(\boldsymbol{z}_{c,k,j} - \boldsymbol{\mu}_c)^T$$

$$= \frac{1}{N_c-1} \sum_{k=1}^{K} \sum_{j=1}^{N_{c,k}} \boldsymbol{z}_{c,k,j} \boldsymbol{z}_{c,k,j}^T - \frac{N_c}{N_c-1} \boldsymbol{\mu}_c \boldsymbol{\mu}_c^T$$

$$= \sum_{k=1}^{K} \frac{1}{N_c-1} \left( (N_{c,k}-1) \boldsymbol{\Sigma}_{c,k} + N_{c,k} \cdot \boldsymbol{\mu}_{c,k} \boldsymbol{\mu}_{c,k}^T \right) - \frac{N_c}{N_c-1} \boldsymbol{\mu}_c \boldsymbol{\mu}_c^T$$

$$= \sum_{k=1}^{K} \frac{N_{c,k}-1}{N_c-1} \boldsymbol{\Sigma}_{c,k} + \sum_{k=1}^{K} \frac{N_{c,k}}{N_c-1} \boldsymbol{\mu}_{c,k} \boldsymbol{\mu}_{c,k}^T - \frac{N_c}{N_c-1} \boldsymbol{\mu}_c \boldsymbol{\mu}_c^T.$$

# B Details of Centered Kernel Alignment (CKA)

Given the same input data, Centered Kernel Alignment (CKA) [23] compute the similarity of the output features between two different neural networks. Let $N$ denote the size of the selected data set $D_{cka}$, and $d_1$ and $d_2$ denote the dimension of the output feature of the two networks respectively. $D_{cka}$ is used for extracting features matrix $Z_1 \in \mathbb{R}^{N \times d_1}$ from one representation network and feature matrix $Z_2 \in \mathbb{R}^{N \times d_2}$ from another representation network. Then the two representation matrices are pre-processed by centering the columns. The linear CKA similarity between two representations X and Y can be computed as below:

$$CKA(X,Y) = \frac{\|X^T Y\|_F^2}{\|X^T X\|_F^2 \|Y^T Y\|_F^2}.$$

In our experiments, we adopt linear CKA to measure the similarity between different local models during federated training. We call the global model optimized on the client's local data for 10 epochs as 'local model'. $d_1$ and $d_2$ are both 256. Since we conduct experiments on CIFAR-10, $N$ is 50,000.

# C Privacy Protection

Each raw representation corresponds to a single input sample, so it may easily leak information about the client's single examples. However, if the mean or covariance is computed from only a few samples, would they expose information about the client's single examples?

To answer this question, we have resorted image reconstruction by feature inversion method in [48] to check whether the raw image can be reconstructed by inverting the representation through the pre-trained model parameters. Experiments are conducted on ImageNet [1] with a pre-trained ResNet-50. As shown in Figure 8 and Figure 9, the image recovered from the raw representation is similar to the corresponding raw image. One can generally identify the category of object. By contrast, the image recovered from the Gaussian mean computed by only 3 samples looks largely different from the user's raw images. It's hard to tell the objects in the recovered images. In conclusion, transmitting per-client Gaussian statistics is basically privacy-preserving when facing feature inversion attack.

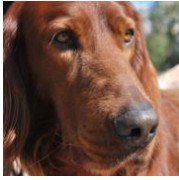 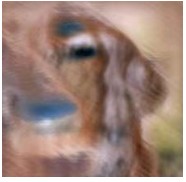

Raw image      Reconstructed image

Figure 8: Image reconstructed from raw feature.

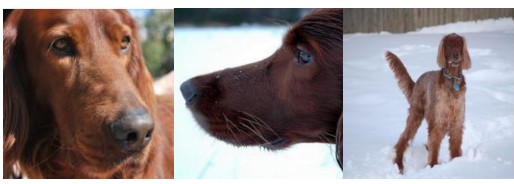 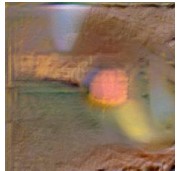

Raw images      Reconstructed image

Figure 9: Image reconstructed from mean of raw features.

## D    Extra Experimental Results

### D.1    How does CCVR improve the classifier?

To check whether CCVR can eliminate the classifier's bias, we visualization the L2-norms of the classifier weight vectors before and after applying CCVR on CIFAR-10 with the concentration parameter $\alpha$ set as $0.1$. As shown in Figure 10, the distributions of the classifier weight L2-norms of FedAvg, FedProx, and MOON are imbalanced without CCVR. However, the imbalanceness is observably alleviated after applying CCVR. It means that the classifiers become fairer when making decisions, considering a larger L2-norm often yields a larger logit, and thus a higher possibility of being predicted.

### D.2    Why not choose regularization during training instead of post-calibration?

In Section 3.3, we observe that regularizing the classifier (either the parameter or the L2-norm) during training can only improve little in most cases. To understand the reason behind these minor improvements, we visualize the means of the CKA similarities of different layers of different federated training algorithms on CIFAR-10 with the concentration parameter $\alpha$ set as $0.1$. From Figure 11, we can see that the two methods (FedProx and MOON) which surpass FedAvg enhance the CKA similarity across all layers over FedAvg. This indicates that the local models trained with FedProx and MOON suffer less from client drift, both on their classifiers and representations. However, we can also observe that if we restrict the weights of the classifier by either clsnorm or clsprox mentioned in Section 3.3, the feature similarities of certain layers of different local models are reduced. Moreover, these restrictions seem too strict for the classifier, making the features outputted by different local classifiers less similar. To conclude, regularization during training not only affects the classifier, but also the feature extractor. In other words, it may deteriorate the quality of learned representation since the classifier learning and representation learning are not fully decoupled. In that case, adopting a classifier post-calibration technique would be a wiser choice.

### D.3    Comparison of the effectiveness of CCVR on FedAvg, FedProx and MOON.

From Table 2, we can observe that the improvement brought by CCVR is less prominent on MOON compared with FedAvg and FedProx for CINIC-10 (3.75% v.s. 9.79% and 9.53%). To understand why it happens, we now

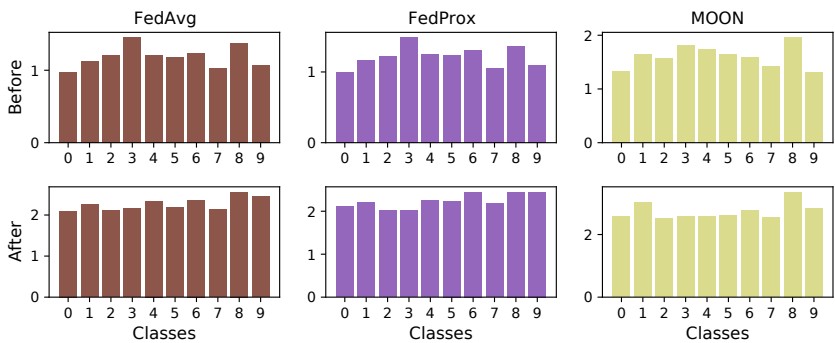

Figure 10: The classifer weight L2-norm of FedAvg, FedProx and MOON before and after applying CCVR on CIFAR-10.

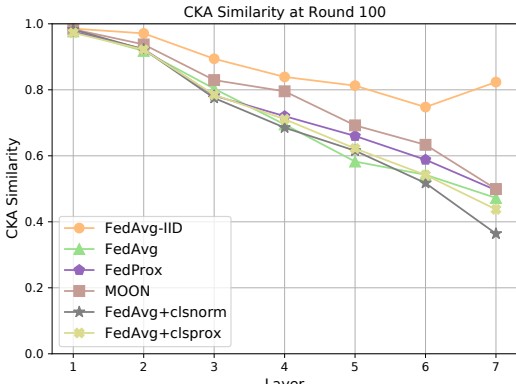

Figure 11: The means of the CKA similarities of different layers for different methods on CIFAR-10.

provide additional visualization results. We first take a closer look at the L2-norm of the classifier weight trained with FedAvg, FedProx, and MOON. As shown in Figure 12, different from that of FedAvg and FedProx, the L2-norm distribution of the classifier trained with MOON is not related to the label distribution at the beginning of the federated training (Round 1). Moreover, the classifier trained with MOON tends to be biased to different classes from FedAvg and FedProx at the end of federated training (Round 100). This implies that MOON's regularization of the representation affects the classifier in an underlying manner.

We now further analyze the representations learned by FedAvg, FedProx, and MOON. Figure 13 demonstrates the t-SNE visualization of the features learned by FedAvg, FedProx, and MOON. We can observe that MOON encourages the model to learn low-entropy feature clusters (high intra-class compactness). Meanwhile, the decision margin becomes larger, bringing more tolerance to the classifier. In other words, the feature space is more discriminative. Though the classifier may have certain biases, the number of misclassifications would also be reduced. As a result, CCVR is left with much less room for improvement.

### D.4 How many virtual features to generate?

In Section 5.4, we study the effect of the number of virtual features $M_c$ when applying CCVR to FedAvg. We now provide additional results of applying CCVR with a different number of virtual features $M_c$ to FedProx and MOON. From Figure 14 and Table 3, we can get similar conclusions to that in Section 5.4: larger $M_c$ yields higher accuracy when faced with highly heterogeneous distributions ($\alpha = 0.05$ and $\alpha = 0.1$), but $M_c$ is more sensitive when faced with a more balanced distribution. Moreover, we observe that the optimal $M_c$s for FedAvg, FedProx and MOON on CIFAR-10 with $\alpha = 0.5$ are 2000, 1000 and 100 respectively. It indicates that MOON seems to require fewer virtual features for calibration compared with FedAvg and FedProx when faced with a more uniform distribution.

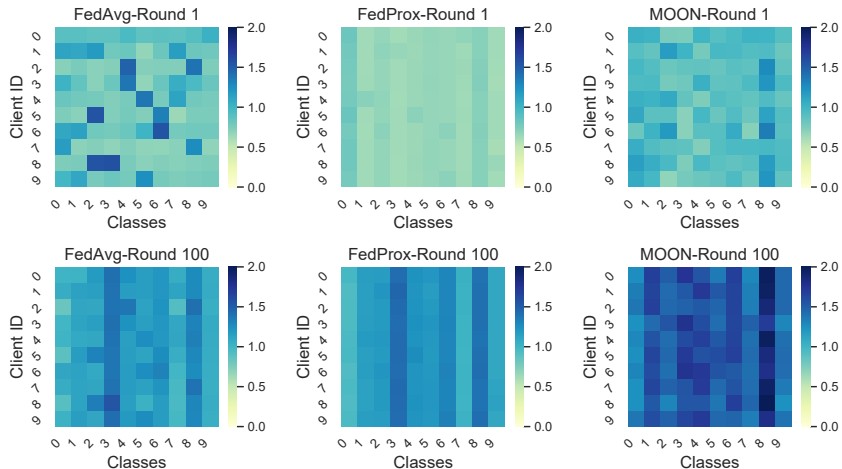

Figure 12: The L2-norm distribution of the classifier across clients of FedAvg, FedProx and MOON in different rounds on CIFAR-10 with $\alpha = 0.1$.

Table 3: Accuracy@1 (%) of CCVR on CIFAR-10 with different numbers of virtual features per class.

| | # virtual features | $\alpha = 0.5$ | $\alpha = 0.1$ | $\alpha = 0.05$ |
|---|---|---|---|---|
| FedAvg(Before Calibration) | - | 68.62±0.77 | 58.55±0.98 | 52.33±0.43 |
| CCVR (Ours.) | 50 | 70.99±0.98 (↑2.37) | 61.07±0.42 (↑2.52) | 54.24±0.83 (↑1.91) |
| | 100 | 71.03±0.40 (↑2.41) | 61.43±0.96 (↑2.88) | 54.48±0.77 (↑2.15) |
| | 500 | 70.85±0.83 (↑2.23) | 62.29±0.73 (↑3.74) | 54.63±1.07 (↑2.30) |
| | 1000 | 70.71±0.71 (↑2.09) | 62.59±0.69 (↑4.04) | 55.03±0.86 (↑2.70) |
| | 2000 | 71.33±0.73 (↑2.71) | 62.68±0.54 (↑4.13) | 54.95±0.61 (↑2.62) |
| FedProx(Before Calibration) | - | 69.07±1.07 | 58.93±0.64 | 53.00±0.32 |
| CCVR (Ours.) | 50 | 70.72±1.02 (↑1.65) | 61.34±0.30 (↑2.41) | 54.78±0.99 (↑1.78) |
| | 100 | 70.99±1.21 (↑1.92) | 61.89±0.40 (↑2.96) | 55.01±1.07 (↑2.01) |
| | 500 | 70.94±0.94 (↑1.87) | 62.29±0.44 (↑3.36) | 55.54±0.97 (↑2.54) |
| | 1000 | 71.16±1.14 (↑2.09) | 62.59±0.49 (↑3.66) | 55.55±0.82 (↑2.55) |
| | 2000 | 70.81±0.84 (↑1.74) | 62.60±0.43 (↑3.67) | 55.79±1.07 (↑2.79) |
| MOON(Before Calibration) | - | 70.48±0.36 | 57.36±0.85 | 49.91±0.38 |
| CCVR (Ours.) | 50 | 71.16±0.18 (↑0.68) | 61.36±0.44 (↑4.00) | 53.36±0.52 (↑3.45) |
| | 100 | 71.29±0.11 (↑0.81) | 62.17±0.74 (↑4.81) | 54.09±0.96 (↑4.18) |
| | 500 | 71.22±0.19 (↑0.74) | 62.10±0.45 (↑4.74) | 55.02±0.75 (↑5.11) |
| | 1000 | 71.11±0.11 (↑0.63) | 62.79±0.85 (↑5.43) | 55.61±0.44 (↑5.70) |
| | 2000 | 71.10±0.19 (↑0.62) | 62.22±0.70 (↑4.86) | 55.60±0.63 (↑5.69) |

## D.5 Does different number of clients affect CCVR's performance?

We conduct additional experiments on CIFAR-10 ($\alpha = 0.1$) with different numbers of clients $N \in \{10, 50, 100\}$. From Table 4, we can observe that CCVR steadily improves accuracy for all the methods in all settings. To conclude, varing number of clients does not affect CCVR's effectiveness.

## D.6 Full results of classifier calibration with whole data and partial data.

In Table 5, we provide full results of classifier calibration for FedAvg and FedProx with whole data and partial data. Note that these results are corresponding to Figure 4.

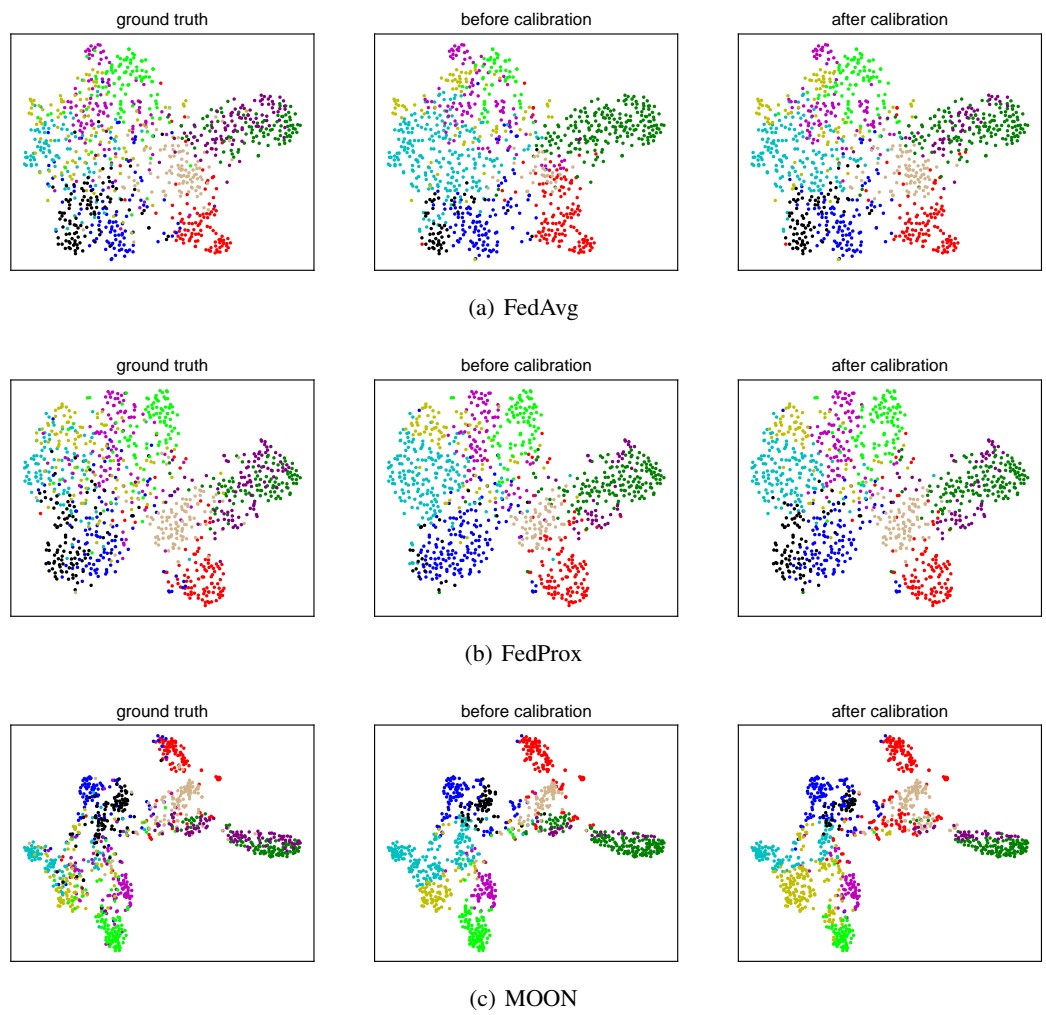

Figure 13: t-SNE visualization of the features learned by FedAvg, FedProx and MOON on CINIC-10. The features are colored by the ground truth and the predictions of the classifier before and after applying CCVR. Best viewed in color.

# E    Experimental Details

## E.1    Datasets

In Figure 15, we visualize the label distributions among the training sets of a population of non-identical clients. As we can see, the label distributions are quite heterogeneous. Specifically, for non-IID partition strategy, the number of samples varies for each client, and each client may have only a few categories of samples. There are 10 clients for all the datasets. The concentration parameter $\alpha$ is set to be 0.5 for CIFAR-100 and CINIC-10. To make fair comparisons between different methods, the data distributions are fixed in our experiments.

## E.2    Model Architectures

Table 6 shows the details of the simple convolutional neural network used for CIFAR-10. Note that it's the same with the model architecture used in [8]. Table 7 provides the details of the MobileNetV2 [43] used for CIFAR-100 and CINIC-10. For CIFAR-100, we change the output dimension of the classifier to 100.

## E.3    Hyperparameters

We summarize all the hyperparameters used in our experiments in Table 8. All the experiments are repeated with three different random seeds.

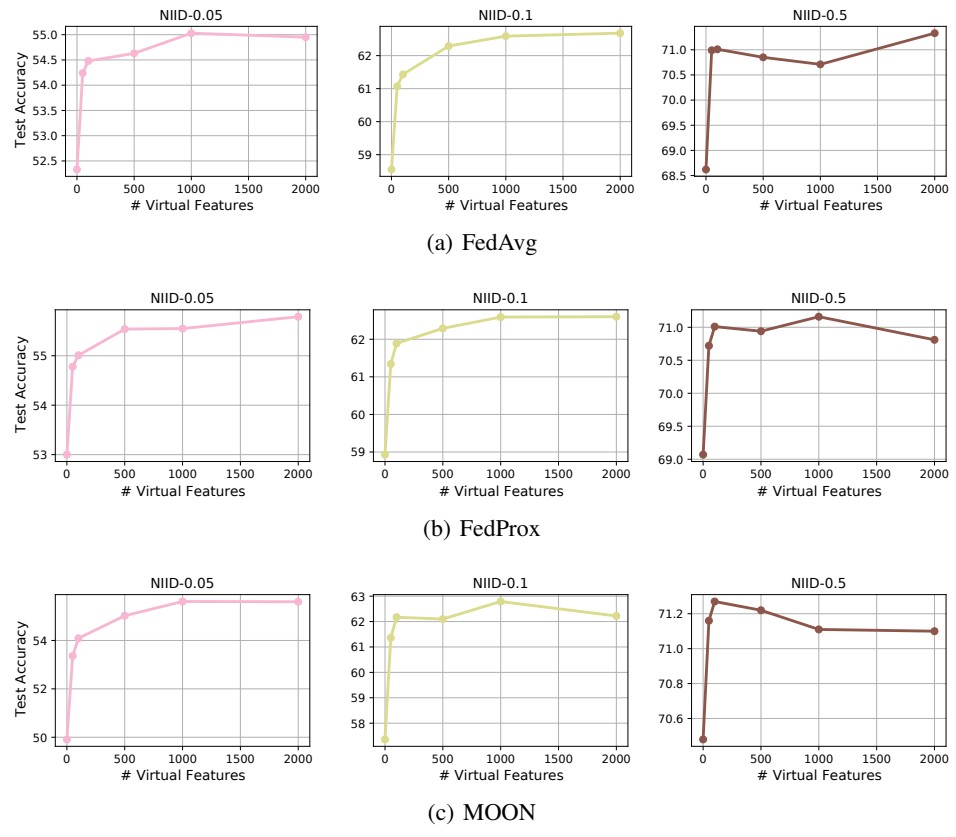

Figure 14: Accuracy@1 (%) of CCVR on CIFAR-10 with different numbers of virtual features.

Table 4: Accuracy@1 (%) on CIFAR-10 ($\alpha = 0.1$) with a varying number of clients $N$.

| | Method | $N = 10$ | $N = 50$ | $N = 100$ |
|---|---|---|---|---|
| | FedAvg | 58.55 | 57.94 | 55.42 |
| No Calibration | FedProx | 58.93 | 58.49 | 55.85 |
| | MOON | 57.36 | 58.51 | 56.26 |
| | FedAvg | 62.68 (↑ 4.13) | 61.89 (↑ 3.95) | 59.19 (↑ 3.77) |
| CCVR | FedProx | 62.60 (↑ 3.67) | 61.69 (↑ 3.20) | 59.04 (↑ 3.19) |
| | MOON | 62.22 (↑ 4.86) | 61.63 (↑ 3.12) | 59.49 (↑ 3.23) |

Table 5: Accuracy@1 (%) on CIFAR-10 of classifier calibration with whole data and partial data.

| Method | $\alpha = 0.5$ | $\alpha = 0.1$ | $\alpha = 0.05$ |
|---|---|---|---|
| FedAvg(Before Calibration) | 68.62±0.77 | 58.55±0.98 | 52.33±0.43 |
| Whole Data | 72.51±0.53 (↑ 3.89) | 64.70±0.94 (↑ 6.15) | 57.53±1.00 (↑ 5.20) |
| Partial Data (1000 samples per class) | 72.30±0.50 (↑ 3.68) | 64.55±1.05 (↑ 6.00) | 56.86±1.01 (↑ 4.53) |
| Partial Data (100 samples per class) | 72.06±0.47 (↑ 3.44) | 63.58±1.22 (↑ 5.03) | 55.65±0.83 (↑ 3.32) |
| FedProx(Before Calibration) | 69.07±1.07 | 58.93±0.64 | 53.00±0.32 |
| Whole Data | 72.26±1.22 (↑ 3.19) | 64.63±0.93 (↑ 5.70) | 57.33±0.72 (↑ 4.33) |
| Partial Data (1000 samples per class) | 72.09±1.15 (↑ 3.02) | 64.14±1.00 (↑ 5.21) | 56.85±0.88 (↑ 3.85) |
| Partial Data (100 samples per class) | 71.90±1.07 (↑ 2.83) | 63.21±0.92 (↑ 4.28) | 55.63±0.72 (↑ 2.63) |

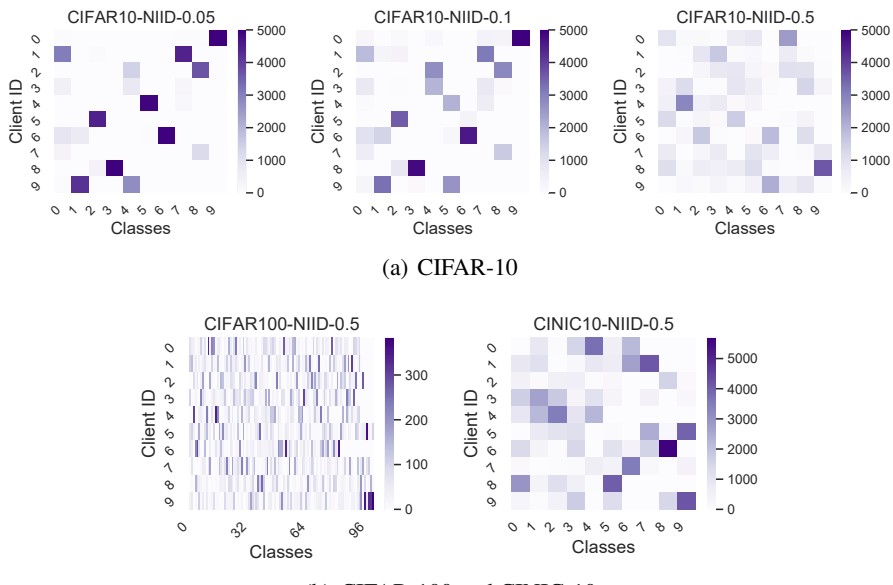

(a) CIFAR-10

(b) CIFAR-100 and CINIC-10

Figure 15: Label distributions of CIFAR-10, CIFAR-100, and CINIC-10 across the clients.

Table 6: Detailed information of the simple convolutional neural network used for CIFAR-10. For convolution layer (Conv2d), the parameters are listed with a sequence of input channel, output channel, kernel size and stride. For max pooling layer (MaxPool2d), we list the kernel size. For fully connected layer (Linear), we list the input dimension and the output dimension.

| Layer | Details | Repetition |
|---|---|---|
| layer 1 | Conv2d(3, 6, k=(5, 5), s=(1, 1))
ReLU()
MaxPool2d(k=(2, 2)) | ×1 |
| layer 2 | Conv2d(6, 16, k=(5, 5), s=(1, 1))
ReLU()
MaxPool2d(k=(2, 2)) | ×1 |
| layer 3 | Linear(400, 120)
ReLU() | ×1 |
| layer 4 | Linear(120, 84)
ReLU() | ×1 |
| layer 5 | Linear(84, 84)
ReLU() | ×1 |
| layer 6 | Linear(84, 256) | ×1 |
| layer 7 (Classifier) | Linear(256, 10) | ×1 |

Table 7: Detailed information of the MobileNetV2 used for CIFAR-100 and CINIC-10. For convolution layer (Conv2d), the parameters are listed with a sequence of input channel, output channel, kernel size, stride and padding. Note that the parameter "g" represents that the corresponding layer is a depthwise convolution. For average pooling layer (AvgPool2d), we list the kernel size. For fully connected layer (Linear), we list the input dimension and the output dimension. There are skip connections in the bottlenecks where the input channels equals to the output channels and the stride of the first convolution layer equals 1. Note that the output dimension of the classifier is replaced with 100 for CIFAR-100.

| Block | Details | Repetition |
|---|---|---|
| | Conv2d(3, 32, k=(3, 3), s=(1, 1), pad=(1, 1)) | ×1 |
| block 1 | Conv2d(32, 32, k=(3, 3), s=(1, 1), pad=(1, 1), g=48)
Conv2d(32, 16, k=(1, 1), s=(1, 1)) | ×1 |
| block 2 | Conv2d(16, 96, k=(1, 1), s=(1, 1))
Conv2d(96, 96, k=(3, 3), s=(1, 1), pad=(1, 1), g=144)
Conv2d(96, 24, k=(1, 1), s=(1, 1)) | ×1 |
| | Conv2d(24, 144, k=(1, 1), s=(1, 1))
Conv2d(144, 144, k=(3, 3), s=(1, 1), pad=(1, 1), g=240)
Conv2d(144, 24, k=(1, 1), s=(1, 1)) | ×1 |
| block 3 | Conv2d(24, 144, k=(1, 1), s=(1, 1))
Conv2d(144, 144, k=(3, 3), s=(2, 2), pad=(1, 1), g=240)
Conv2d(144, 32, k=(1, 1), s=(1, 1)) | ×1 |
| | Conv2d(32, 192, k=(1, 1), s=(1, 1))
Conv2d(192, 192, k=(3, 3), s=(1, 1), pad=(1, 1), g=288)
Conv2d(192, 32, k=(1, 1), s=(1, 1)) | ×2 |
| block 4 | Conv2d(32, 192, k=(1, 1), s=(1, 1))
Conv2d(192, 192, k=(3, 3), s=(2, 2), pad=(1, 1), g=288)
Conv2d(192, 64, k=(1, 1), s=(1, 1)) | ×1 |
| | Conv2d(64, 384, k=(1, 1), s=(1, 1))
Conv2d(384, 384, k=(3, 3), s=(1, 1), pad=(1, 1), g=576)
Conv2d(384, 64, k=(1, 1), s=(1, 1)) | ×3 |
| block 5 | Conv2d(64, 384, k=(1, 1), s=(1, 1))
Conv2d(384, 384, k=(3, 3), s=(1, 1), pad=(1, 1), g=576)
Conv2d(384, 96, k=(1, 1), s=(1, 1)) | ×1 |
| | Conv2d(96, 576, k=(1, 1), s=(1, 1))
Conv2d(576, 576, k=(3, 3), s=(1, 1), pad=(1, 1), g=864)
Conv2d(576, 96, k=(1, 1), s=(1, 1)) | ×2 |
| block 6 | Conv2d(96, 576, k=(1, 1), s=(1, 1))
Conv2d(576, 576, k=(3, 3), s=(2, 2), pad=(1, 1), g=864)
Conv2d(576, 160, k=(1, 1), s=(1, 1)) | ×1 |
| | Conv2d(160, 960, k=(1, 1), s=(1, 1))
Conv2d(960, 960, k=(3, 3), s=(1, 1), pad=(1, 1), g=1440)
Conv2d(960, 160, k=(1, 1), s=(1, 1)) | ×2 |
| block 7 | Conv2d(160, 960, k=(1, 1), s=(1, 1))
Conv2d(960, 960, k=(3, 3), s=(1, 1), pad=(1, 1), g=1440)
Conv2d(960, 320, k=(1, 1), s=(1, 1)) | ×1 |
| | Conv2d(320, 1280, k=(1, 1), s=(1, 1))
AvgPool2d(k=(4, 4)) | ×1
×1 |
| Classifier | Linear(1280, 10) | ×1 |

Table 8: Hyperparameters used in our experiments.

| Methods | Hyperparameters | CIFAR-10-0.05 | CIFAR-10-0.1 | CIFAR-10-0.5 | CIFAR-100 | CINIC-10 |
|---|---|---|---|---|---|---|
| FedAvg/FedAvgM | communication rounds | | | 100 | | |
| | optimizer | | | SGD | | |
| | learning rate | | | 0.01 | | |
| | weight decay | | | 1e-5 | | |
| | momentum | | | 0.9 | | |
| | local epoch | | | 10 | | |
| | clients per round | | | 10 | | |
| | batch size | 64 | 64 | 64 | 64 | 512 |
| clsprox | $\mu$ | 0.01 | 0.001 | 0.001 | - | - |
| FedProx | communication rounds | | | 100 | | |
| | optimizer | | | SGD | | |
| | learning rate | | | 0.01 | | |
| | weight decay | | | 1e-5 | | |
| | momentum | | | 0.9 | | |
| | local epoch | | | 10 | | |
| | clients per round | | | 10 | | |
| | batch size | 64 | 64 | 64 | 64 | 512 |
| | $\mu$ | 0.01 | 0.001 | 0.001 | 0.001 | 0.01 |
| MOON | contrastive temperature | | | 0.5 | | |
| | optimizer | | | SGD | | |
| | learning rate | | | 0.01 | | |
| | weight decay | | | 1e-5 | | |
| | momentum | | | 0.9 | | |
| | epoch | | | 10 | | |
| | clients per round | | | 10 | | |
| | batch size | 64 | 64 | 64 | 64 | 512 |
| | $\mu$ | 5 | 1 | 1 | 1 | 1 |
| FedAvg + CCVR | optimizer | | | SGD | | |
| | weight decay | | | 1e-5 | | |
| | momentum | | | 0.9 | | |
| | batch size | | | 64 | | |
| | epoch | 10 | 10 | 30 | 30 | 50 |
| | number of virtual features per class | 2000 | 2000 | 100 | 500 | 1000 |
| | learning rate | 0.001 | 0.001 | 0.001 | 1e-5 | 0.001 |
| FedProx + CCVR | optimizer | | | SGD | | |
| | weight decay | | | 1e-5 | | |
| | momentum | | | 0.9 | | |
| | batch size | | | 64 | | |
| | epoch | 10 | 10 | 30 | 30 | 50 |
| | number of virtual features per class | 2000 | 2000 | 100 | 500 | 1000 |
| | learning rate | 0.001 | 0.001 | 0.001 | 1e-5 | 0.001 |
| FedAvgM + CCVR | optimizer | | | SGD | | |
| | weight decay | | | 1e-5 | | |
| | momentum | | | 0.9 | | |
| | batch size | | | 64 | | |
| | epoch | 10 | 10 | 50 | 10 | 50 |
| | number of virtual features per class | 2000 | 2000 | 100 | 500 | 1000 |
| | learning rate | 0.001 | 0.001 | 0.001 | 1e-5 | 0.001 |
| MOON + CCVR | optimizer | | | SGD | | |
| | weight decay | | | 1e-5 | | |
| | momentum | | | 0.9 | | |
| | batch size | | | 64 | | |
| | epoch | 10 | 10 | 30 | 10 | 10 |
| | number of virtual features per class | 2000 | 2000 | 100 | 500 | 1000 |
| | learning rate | 0.001 | 0.001 | 0.001 | 1e-5 | 0.001 |
| Whole Data | optimizer | | | SGD | | |
| | learning rate | | | 0.001 | | |
| | weight decay | | | 1e-5 | | |
| | momentum | | | 0.9 | | |
| | batch size | | | 64 | | |
| | epoch | | | 50 | | |