# OpenReview forum: "No Fear of Heterogeneity: Classifier Calibration for Federated Learning with Non-IID Data"
_NeurIPS.cc/2021/Conference — NeurIPS 2021 Poster_

### Official Review · Reviewer_JNfg · 2021-06-25

**Rating:** 6
**Confidence:** 4

**Summary:**

This paper claims that current federated learning (FL) systems suffer from performance reduction due to high variance in the features learned by the final layers of the network across clients. The authors motivate their approach by training a local model per client and observing the representations learned by each layer. They show ​that the first layers generate similar representations across clients while the representations of more advanced layers are less similar. Therefore, they propose a simple post-processing step to correct the weights of the classification layer of the shared model after training with standard FL procedures (such as FedAvg). They assume the representation in the last layer follows a Gaussian distribution per class.  At the end of the training stage, they store statistics of the representations in the last layer for each client. Based on these statistics, they generate pseudo-data on the server and retrain only the last layer. The final model composed of the initial layers (that were trained regularly with standard FL approaches) and the final layer that was relearned.

**Limitations And Societal Impact:**

The authors addressed the potential limitations and negative societal impact of their work.

**Main Review:**

Overall it is a nice paper; however, I don't think that it is original or significant enough to be accepted to NeurIPS. I am willing to change my mind in light of new information from the authors and by addressing the below comments.

This paper presents the following merits:
- The approach is simple and doesn't impose a large overhead over current learning procedures.
- The approach is general in the sense that it can be integrated with many FL learning procedures.
- In CINIC-10 the improvements are significant over compared baseline methods.
- The experimental setup is clear. Code was provided which is great.
- The analysis of the method is good and exhaustive.

Some points that I found problematic:
- I think that the comparison made in the paper to baseline methods is severely lacking. Therefore, it is hard to accurately quantify how significant the improvements over current methods really are. Many studies were done in the past few years showing that under heterogeneous distribution FedAvg does not perform well (e.g., [1]). A comparison to more recent baselines (even ones that can not be integrated with this method) is needed.
- All the results in the paper are with 10 clients. Real-life federate learning systems have much more than that. I believe that the performance should be tested under a varying amount of clients.
- The results on CIFAR-100 indicate that this method will no perform well when the data per client is limited. This point was also stated by the authors and I appreciate that. However, federated learning is of most use under this scenario exactly, when there is enough data per client one may do local training only. I think that this point weakens the paper's claim.
- Although the analysis made in the paper showing that the representations of later layers are less similar across clients is nice, I think this is not surprising, especially given that there isn't a domain shift across clients. This is in line with the current understanding of NNs that early layers learn general (and transferable) representations while the representation of later layers is more specific [2].

Some minor comments:
- The authors cannot expect the average user to be familiar with Centered Kernel Alignment (CKA) and should provide a detailed description about it in the appendix.
- I think that the colors in the t-SNE figures should be associated with the actual classes (and refer in the text to points associated with that class by the class name, e.g., L297).
- Some references should be altered from arXiv to the publication venue (e.g., [3], [4]).


[1] Arivazhagan, M. G., Aggarwal, V., Singh, A. K., & Choudhary, S. (2019). Federated learning with personalization layers. arXiv preprint arXiv:1912.00818.
[2] Yosinski, J., Clune, J., Bengio, Y., & Lipson, H. (2014, December). How transferable are features in deep neural networks?. In Proceedings of the 27th International Conference on Neural Information Processing Systems-Volume 2 (pp. 3320-3328).
[3] Li, T., Sahu, A. K., Talwalkar, A., & Smith, V. (2020). Federated learning: Challenges, methods, and future directions. IEEE Signal Processing Magazine, 37(3), 50-60.
[4] Wang, H., Yurochkin, M., Sun, Y., Papailiopoulos, D., & Khazaeni, Y. (2020, April). Federated Learning with Matched Averaging. In International Conference on Learning Representations.

**Time Spent Reviewing:**

5

---

> ### Author Response · Authors · 2021-08-10
> **Response to Reviewer JNfg: Clarification on the significance and experiments with additional baselines and datasets.**
>
> Thanks very much for your very detailed and valuable comments! We are glad you appreciate that our proposed approach CCVR is simple, effective, and easy to integrate with most federated learning methods. We understand that your main concern is CCVR’s effectiveness under more real-world settings and CCVR’s superiority compared with more federated learning baselines. We will address them as below.
>
>  **A1.** [*I don't think that it is original or significant enough to be accepted to NeurIPS.*]
> - We would like to clarify that our work is novel and non-trivial.
>    - Existing works in FL on non-IID data lack a deep understanding of how the data heterogeneity affects each layer of a deep classification model. Our work bridges this gap by performing **the first systematic empirical study** on the hidden representations learned by different layers.
>    - The two findings from the empirical study are insightful: (1) there exists a greater bias in the classifier than other layers, and (2) the classification performance can be significantly improved by post-calibrating the classifier after federated training. The two insights may motivate future works in the FL community, as appreciated by Reviewer B69V.
>   - Our **simple yet effective** method CCVR addresses the performance degradation problem of FL on non-IID data from **a novel perspective**: classifier debiasing, which has never been discussed in previous works. We have gone through an extensive literature review and found that existing research on FL with non-IID data has generally pursued the four directions: client drift mitigation, aggregation scheme, data sharing, and personalized federated learning. Strictly speaking, our proposed CCVR does not fall into any of the four research directions.
>   - Though the idea of using Gaussian statistics to calibrate federated learning models seems simple, our empirical results show prominent accuracy gains. As acknowledged by Reviewer 445Q, **‘the field of FL generally lacks this type of insightful yet powerful research’**.
>   - Different from most existing works, our CCVR is a post-hoc method and does not require any modification to the existing FL training procedure. Our method is general in the sense that **it can be integrated with many FL learning procedures to get a free lunch of accuracy gains**.
>
>
> **A2.** [*Lacking comparisons to more recent baselines.*]
> - To the best of our knowledge, one of our considered baselines MOON (published at CVPR 2021) is the SOTA method for federated learning with non-IID data. However, we totally understand that the reviewer would like to see more comprehensive comparisons between our method and other baselines. So, we here introduce additional baselines FedPer[1] (suggested by the reviewer) and FedAvgM[2] (a recent popular algorithm that considers momentum update on the server-side). We note that the original FedPer paper [1] considers the personalization setting where the personalized models are tested on local testing sets of clients. However, our proposed CCVR is designed for the classic FL setting where a global model is trained for all users and tested on a stand-alone global testing set.  Moreover, FedPer cannot be combined with CCVR. To fairly compare FedPer[1] and CCVR, we follow the evaluation scheme of personalization setting for all the methods. Below are the results on CIFAR-10 with $\alpha=0.1$. We can observe that **CCVR consistently improves the baseline methods**. Moreover, the performance of the recent FL algorithm + CCVR outperforms FedPer. Full results on other datasets will be included in the revision.
>
> |  Method   | Before Calibration (%)  | CCVR (%) |
> |  ----  | :----: | :----: |
> |  FedAvg |     58.55    |     62.68 (+4.13) |
> |  FedProx |    58.93   |      62.60 (+3.67) |
> |  MOON |    57.36  |      62.22 (+4.86)|
> |  FedAvgM [2] |    61.02   |      **63.43** (+2.41) |
> |  FedPer [1] |    61.37  |  - |
>
>
> **A3.** [*The performance should be tested under a varying amount of clients.*]
> - Thank you for pointing it out. We conduct additional experiments on CIFAR-10 ($\alpha=0.1$) with different numbers of clients N.  From the Table below, we can observe that CCVR steadily improves accuracy for all the methods in all settings.
>
> |  Method | $N=10$ (%)  | $N=50$ (%) | $N=100$ (%)  |
> |  ----  | :----: | :----: | :----: |
> |  FedAvg |     58.55    |     57.94 |  55.42  |
> |  FedProx |    58.93   |      58.49 |   55.85   |
> |  MOON |    57.36  |      58.51 |  56.26 |
> |  FedAvg + CCVR |     **62.68** (+4.13)    |     **61.89** (+3.95) |   59.19 (+3.77)  |
> |  FedProx + CCVR |    62.60 (+3.67)    |      61.69 (+3.20) |   59.04 (+3.19)  |
> |  MOON + CCVR |    62.22 (+4.86)  |      61.63 (+3.12) |  **59.49** (+3.23) |
>
> - We also conducted additional experiments on a more real-world dataset iNaturalist-User 120K proposed by [2]. iNaturalist-User 120K has 1203 classes, 9275 clients, and 120300 training examples. Since [2] does not provide the original training code, we set the training configurations on our own. We use the same model architecture (MobileNet-v2 with Group Normalization pre-trained on ImageNet) with [2]. Due to the limited time and computation resources, we only train the model for 2500 rounds and only experiment with FedAvg. We randomly select 100 clients per round. $M_c$ of CCVR is set to be 100. Since we didn't adopt the FedVC trick proposed in [2], the results of our re-implementation cannot be directly compared with the results reported in [2].  As shown in the table below, our vanilla FedAvg achieves 43.8% testing accuracy. The accuracy is further increased by incorporating CCVR. It indicates that **CCVR still works well with the real-world setting where the total number of clients is large**. Full results on iNaturalist-User 120K will be included in the revision.
>
> |     | Before Calibration (%)  | CCVR (%)  |
> |  ----  | :----: | :----: |
> |  FedAvg |     43.8    |    **44.5**  |
>
>
> **A4.** [*This method may not perform well when the data per client is limited.*]
> - In iNaturalist-User 120K, there are 9275 clients and 120300 training examples. There is an average of 12.97 samples per user. Over 7000 users have a local sample size of less than 10. From the above result on iNaturalist-User 120K, we can see that **CCVR works well for the setting where the data per client is limited.**
> - Results on CIFAR-100 do not weaken our claim. For CIFAR-100, there is an average of 5000 samples per client. So the subtle improvement on CIFAR-100 is not due to the limited data size of clients. As explained by the first paragraph of Section 5.3, the marginal improvement on CIFAR-100 is mainly due to the low separability of the representations.
> We will provide a detailed analysis of the performance guarantee of our CCVR in the revision. Intuitively, the separability of Gaussian mixtures reflects the separability of learned representations. As we know, the separability of learned representations is positively correlated with the accuracy upper bound of classifier calibration. So, the separability of Gaussian mixtures may be also positively correlated with the accuracy upper bound of classifier calibration. To quantify the separability of GMM, one can resort to [3] to calculate the Overlap Rate (OLR) of GMM. We will provide a rigorous analysis of the relationship between the OLR of GMM and the accuracy upper bound of classifier calibration.
>
> **A5.** [*Although the analysis … is nice, I think this is not surprising.*]
> - We acknowledge the viewpoint that "the early layers of NN learn general (and transferable) representations while the representation of later layers is more specific" may not be surprising in deep learning. However, we still believe that it provides a new perspective of understanding federated learning with heterogeneous data and can motivate future works in the FL community. This is also acknowledged by Reviewer B69V.
> - It is not the main selling point of our work. Another observation in our work is much more interesting: the classification performance can be significantly improved by post-calibrating the classifier after federated training. This observation directly motivates our simple yet effective method.
>
> **A6.** [*Centered Kernel Alignment (CKA).*]
> - Thanks for reminding us. We will provide a detailed description of CKA in the revised appendix.
>
> **A7.** [*t-SNE figures.*]
> - Thanks for the useful suggestion. We will modify the t-SNE figures to improve their readability in the revision.
>
> **A8.** [*Some references are not the most recent version.*]
> - We will modify them in the revision.
>
> [1] Federated learning with personalization layers @Arxiv 2019
>
> [2] Federated visual classification with real-world data distribution @ECCV 2020
>
> [3] Measuring the component overlapping in the Gaussian mixture model @Data Mining and Knowledge Discovery 2011

---

> > ### Comment · Reviewer_JNfg · 2021-08-18
> > **Response to Author's Rebuttal**
> >
> > I would like to thank the authors for the comprehensive response. In light of it, I decided to raise my score from 4 to 6.
> > I still think that a comparison to more *recent* and *strong* baselines should have been done. I pointed to FedPer just to emphasize that even this relatively old and simple baseline outperforms FedAvg. Nevertheless, I was convinced that this paper should pass since it presents a novel research direction and the comparisons now are more reasonable.

---

> > > ### Author Response · Authors · 2021-08-18
> > > **Thanks for Reviewer JNfg's response.**
> > >
> > > Thanks very much for your kind reconsideration! We are very happy that you recognize that our work is significant since it presents a novel research direction in FL, and our new experimental results are promising. We will follow your suggestions and include our new experimental results in the revision.

---

### Official Review · Reviewer_445Q · 2021-07-16

**Rating:** 6
**Confidence:** 5

**Summary:**

The authors proposed CCVR to post-calibrate federated learning classifiers models. Insights were first drawn from preliminary experiments, and CCVR is then tested atop many popular FL methods to demonstrate its efficacy.

**Limitations And Societal Impact:**

I recommend discussing more limitations should the space allow.

**Main Review:**

The idea of using simple statistics to calibrate federated learning models is great. Especially when the authors were able to show prominent gains with a post-hoc method that does not require modifications to existing training methods. The field of FL generally lacks this type of insightful yet powerful research.

With that said, I am providing some comments below:

- This work is missing a critical reference: [1] addresses client distribution drift on large scale FL datasets. It falls under the first category in L28.

- It seems a bit indirect having to aggregate per-client Gaussian mixtures into a global Gaussian mixture. One might argue that this step removes per-client statistics in exchange for better privacy, but in practice the clients might still need to send the full matrices over to the server for the aggregation. Hence it is reasonable that a server utilizes these statistics for learning/fine-tuning. I wish to either see the authors address this privacy comment, or provide some insights on how per-client CCVR (sampling from per-client Gaussians) compares with global CCVR (sampling from global Gaussian)

- In the case of extremely non-identical client distribution such as when alpha -> 0, some clients might hold very sparse classes, and the mean/covariance would be mostly represented by a few samples. The statistics may expose a lot about the client’s single examples.

- Although this work investigates a general phenomenon about FL, the datasets used (CIFAR/CINIC) are still very basic datasets that were not intended for realistic non-identical setups. Sure the datasets can be synthesized to have non-identical clients, by the conventional Dirichlet assignment neglects the fact that client data pool sizes can be drastically different. See [1]. An additional verification on larger datasets would be very helpful.

- Isn’t section 3.3 simply “data sharing for FL model improvement”? If it is, I am receiving contradicting messages from the Introduction and the analysis in this section.

- It’ll be good to provide a brief analysis on the sensitivity against µ, the regularization coefficient.

- In table 1 clsnorm / clsprox is not clearly described. Consider including the names in section 3.3

- Figure 5 does not convey the message effectively. The embeddings are basically identical across rows and the differences are exhibited in the colors. Consider using confusion matrices.

[1] Federated visual classification with real-world data distribution.

I am happy to increase my rating should the comments be addressed properly.

**Time Spent Reviewing:**

4

---

> ### Author Response · Authors · 2021-08-10
> **Response to Reviewer 445Q: Clarification for global CCVR and experiments on a larger dataset [1].**
>
> Thanks very much for your very valuable and constructive review! Particularly, we are happy that you recognize that our work is an insightful yet powerful type of work, which is lacking in the FL community. We now address your main concerns as follows.
>
> **A1**. [*Missing a critical reference: [1].*]
> - Thanks for reminding us. We note that the methods proposed by [1] originate from simple ideas but achieve excellent performance. This is the very merit we advocate when we design our method CCVR. We will discuss it in our revised manuscript.
>
> **A2**. [*It seems a bit indirect having to aggregate per-client Gaussian mixtures into a global Gaussian mixture.*]
> - The reason we consider aggregating per-client Gaussian mixtures into a global Gaussian mixture is to model the global distribution of the representation more accurately.  Moreover, the virtual representation sampled from the Gaussian distribution of each class can be more representative. For example, if a client only has 2 samples of the cat category, then sampling from the Gaussian distribution computed from only 2 samples may be unreliable. A more reasonable way would be to sample from the global Gaussian distribution computed from statistics of other clients with cat images. We have conducted additional experiments on CIFAR-10 ($\alpha=0.1$) to compare per-client CCVR (sampling from per-client Gaussians) and global CCVR (sampling from global Gaussians).  In per-client CCVR, the number of virtual representation M_c of class c is positively correlated with the number of raw representations that are used to compute the Gaussian statistics. Below are the results. We obverse that the accuracy improvement brought by per-client CCVR is lower than that of the global CCVR. **This indicates that the quality of virtual representation sampled by per-client CCVR is lower than that by global CCVR**.
>
> |  Method   | Before Calibration (%)  | Global CCVR (%)  | Per-client CCVR (%) |
> |  ----  | :----: | :----: | :----: |
> |  FedAvg |     58.55    |     **62.68** (+4.13) | **62.19** (+3.64)|
> |  FedProx |    **58.93**   |      62.60 (+3.67) |  61.78 (+2.85)|
> |  MOON |    57.36  |      62.22 (+4.86) |  61.81 (+4.45)|
>
> **A3**. [*Extreme non-identical client distribution case.*]
> - CCVR protects privacy at the basic level because each client only uploads their local gaussian statistics rather than the raw representations. Each raw representation corresponds to a single input sample, so it may easily leak information about the client’s single examples. However, if the mean/covariance is computed from only a few samples, would they expose information about the client’s single examples? Honestly, it is a difficult problem and hard to be addressed in a short period of rebuttal. Its variant is also plaguing the FL community for a long time [2]. We believe this is a very interesting future work to explore. To answer this question, we have resorted to a popular model attack method [3]. We aim to check whether the raw image can be reconstructed by inverting the representation through the pre-trained model parameters. We now share our initial findings: 1) The image recovered from the raw representation is similar to the corresponding raw images. One can generally identify the category of object. 2) The images recovered from the Gaussian mean computed by a few samples (3 samples in our experiment) looks largely different from the user’s raw images. It’s hard to tell the objects in the recovered images. In conclusion, our experiments show that **transmitting per-client statistics is basically privacy-preserving when facing attack methods like [3]**. The visual results and a detailed analysis will be included in the revision.
>
> **A4**. [*An additional verification on larger datasets would be very helpful.*]
> - We have conduct additional experiments on a more real-world dataset iNaturalist-User 120K proposed by [1]. iNaturalist-User 120K has 1203 classes, 9275 clients, and 120300 training examples. Since [1] does not provide the original training code, we set the training configurations on our own. We use the same model architecture (MobileNet-v2 with Group Normalization pre-trained on ImageNet) with [1]. Due to the limited time and computation resources, we only train the model for 2500 rounds and only experiment with FedAvg. We randomly select 100 clients per round. $M_c$ of CCVR is set to be 100. Since we didn't adopt the FedVC trick proposed in [1], the results of our re-implementation cannot be directly compared with the results reported in [1].  As shown in the table below, our vanilla FedAvg achieves 43.8% testing accuracy. The accuracy is further increased by incorporating CCVR. **It shows that CCVR still works well with the real-world setting where the client data pool sizes are drastically different**. Full results on iNaturalist-User 120K will be included in the revision.
>
> |     | Before Calibration (%)  | CCVR (%)  |
> |  ----  | :----: | :----: |
> |  FedAvg |     43.8    |    **44.5**  |
>
> **A5**. [*Isn’t section 3.3 simply “data sharing for FL model improvement”?*]
> - No. Section 3.3 provides three solutions to address classifier bias. The first two methods introduce regularization in the training stage, while the last method “Classifier calibration with IID samples” is a post-hoc approach. Our experiments show that the latter is significantly better than the first two approaches. Through Section 3.3, we hope to provide the insight that when we deal with the classifier bias problem in FL, post-processing may be more useful than regularization in the training phase, as it decouples representation learning and classifier. More explanations can be found in appendix B.2.
>
> **A6**. [*Analysis on the sensitivity against $\mu$.*]
> - In Table 1 of our submission, all the results of clsprox are obtained under the best $\mu$. In the following table, we provide the analysis on the sensitivity against $\mu$. We observe that the smaller the concentration parameter, the smaller the optimal regularization coefficient. Full results will be provided in the revised appendix.
>
> |  clsprox ($\mu$) | $\alpha=0.5$  | $\alpha=0.1$ | $\alpha=0.05$  |
> |  ----  | :----: | :----: | :----: |
> |  0.1   |     68.78       |     58.84     |     51.20  |
> |  0.01  |     **68.82**     |       58.36    |      51.84  |
> |  0.001  |     68.10      |       **59.04** |      **52.38** |
>
>
> **A7**. [*clsnorm / clsprox is not clearly described.*]
> - Thank you for pointing it out. We will address it in the revision.
>
> **A8**. [*Figure 5 does not convey the message effectively.*]
> - Thanks for the useful suggestion. We will add confusion matrices in the revision.
>
> **A9**. [*I recommend discussing more limitations should the space allow.*]
> - Thanks for the valuable suggestion. We are happy to add more discussions on the limitation in our future revision.
>
>
> [1] Federated visual classification with real-world data distribution @ECCV 2020
>
> [2] Inverting Gradients - How easy is it to break privacy in federated learning? @NeurIPS 2020
>
> [3] Deep Image Prior @CVPR 2018

---

> ### Author Response · Authors · 2021-08-24
> **Response to Reviewer 445Q: Supplementary Visual Results**
>
> Thanks again for your thoughtful review. We just learned that visual results such as images can be provided through external links during rebuttal. **We now supplement the visual results mentioned in A3 of our original response**. The link is https://drive.google.com/file/d/1mUdq0AxTU8SXp0BwFuzEriHH-EmaaLiH/view?usp=sharing.
>
> We experimented with a popular model reconstruction attack method [1] to check whether the single raw image can be reconstructed by inverting the representation through the pre-trained model parameters. As we can see, the images recovered from the Gaussian mean (calculated from only 3 samples) are very different from the raw images. It indicates that transmitting per-client statistics does not leak the client’s single examples when facing the popular reconstruction attack [1] in the extreme non-IID case.
>
> Thanks again for your patience. We sincerely hope this update help address your concern. If you have further questions about our work, please don't hesitate to post a comment. We will do our best to respond to you before the discussion period ends.
>
> [1] Deep Image Prior @CVPR 2018

---

### Official Review · Reviewer_BVNc · 2021-07-18

**Rating:** 5
**Confidence:** 4

**Summary:**

This paper proposes to bridge the gap between data heterogeneity and layer-wise sensitivity in deep models in Fed learning. The contributions of this paper are as follows: (1) the authors claim to be the first systematic study on hidden representation sensitivity to IID noise, the result of which reveals that the classifier layer is the most sensitive. (2) the authors then propose a calibration procedure that preserves the privacy of inputs while dealing with mitigating data drift.

**Main Review:**

Strengths:
- Reasonable summary of related work, broad synthesis of categories related to heterogeneity mitigation is useful.
- The proposed method is very simple and easy to understand.
- Great reproducibility; supplemental material provides code and data for all experiments.

Weaknesses:
- One of the main selling points is that the authors fit a GMM on the last representation layer before the classifier from which to draw samples for calibration. The authors suggest that the alternative approach of directly using hidden representations from actual samples violates privacy. However, fitting a GMM to the last layer essentially leaks information about the distribution (i.e., the sufficient statistics) from which the client data is generated. In this sense, it is unclear whether the proposed approach actually respects the privacy constraint.
- Weak experimental baselines. Only FedAvg, FedProx, and MOON are used for comparison. No baselines from personalized federated learning or aggregation schemes are included. In particular, a more fair comparison should consider personalized federated approaches such as meta-learning, which is in principle similar to the proposed GMM + calibration approach.
- Limited technical contribution. Why use GMM in favor of other (more powerful) generative models? Section 5.3 mentions the separability of learned representations. If a GMM is used to generate synthetic samples, perhaps a more rigorous analysis can be provided to formalize 5.3 findings (i.e., performance guarantees or limitations based on GMM parameters, etc).

Overall the novelty is not very high and experiments do not make a strong case for a NIPS paper.

**Time Spent Reviewing:**

2

---

> ### Author Response · Authors · 2021-08-10
> **Response to Reviewer BVNc: Clarification for GMM and additional baselines**
>
> Thanks very much for your constructive review. We address the main concerns below.
>
> **A1**. [*Fitting a GMM to the last layer essentially leaks information about the distribution.*]
> - Transmitting hidden representations from actual samples is common in existing FL literature, such as [1]. However, each raw representation corresponds to a single input sample, so it may easily leak information about the client’s single examples. We hope CCVR can protect privacy at the basic level by only uploading the local Gaussian statistics instead of the raw representation. To verify this, we experiment with a popular model attack method [2] to check whether the raw image can be reconstructed by inverting the representation through the pre-trained model parameters. The visual results and detailed analysis will be included in the revision. We now share our initial findings: 1) The image recovered from the raw representation is similar to the corresponding raw image. One can generally identify the type of object. 2) The image recovered from the virtual representation generated by CCVR or gaussian mean looks largely different from the user’s raw images. It’s hard to recognize the objects in the reconstructed images. To conclude, our experiments show that transmitting local distribution statistics is more privacy-preserving than transmitting raw representations when facing attack methods like [2].
> - We would like to clarify that the GMM is only our assumption when modeling the real distribution. Though GMM does not contain full information on the real distribution, it is enough for our classifier calibration task. As shown by Table 2 and Table 3 in our submission, though there is no 100% match between GMM and the real distribution, GMM brings considerable accuracy improvement for classifier calibration in FL.
>
>
>
> **A2**. [*No baselines from personalized federated learning or aggregation schemes are included.*]
> - We disagree with the viewpoint that “personalized federated approaches such as federated meta-learning is in principle similar to the proposed GMM + calibration approach”. The two approaches study different settings of FL. According to [3], there are generally two settings in FL: the **classic setting** where a global model is trained for all users and tested on a stand-alone global testing set (studied by the original FedAvg paper and most of FL literature) and the **personalization setting** where personalized models are trained for each user and tested on local testing sets of clients (studied by most personalized FL works). Our work mainly focuses on the classic setting, while personalized federated approaches focus on the personalization setting. The evaluation methods are different, so they cannot be directly compared.
> - We understand that reviewer would like to see more comprehensive comparisons between our method and baselines from different categories, so we introduce additional baselines FedAvgM[4] (a popular algorithm from aggregation scheme category) and FedPer[5] (a personalization FL algorithm suggested by Reviewer JNfg). Note that FedPer cannot be directly compared with CCVR and cannot be integrated with CCVR. To make a fair comparison with FedPer, we follow the evaluation scheme of personalization setting for all methods: test the global model on the local testing sets of clients. Below are the results on CIFAR-10 with $\alpha=0.1$. We observe that the results of baselines are consistently improved by incorporating CCVR. Moreover, the performance of existing FL methods with CCVR outperforms FedPer. Full results on other datasets will be included in the revision.
>
> |  Method   | Before Calibration (%)  | CCVR (%) |
> |  ----  | :----: | :----: |
> |  FedAvg |     58.55    |     62.68 (+4.13) |
> |  FedProx |    58.93   |      62.60 (+3.67) |
> |  MOON |    57.36  |      62.22 (+4.86)|
> |  FedAvgM [4] |    61.02   |      **63.43** (+2.41) |
> |  FedPer [5] |    61.37  |  - |
>
> **A3**. [*Why use GMM in favor of other (more powerful) generative models?*]
> - We choose GMM mainly because it is simple and effective. This merit is also appreciated by Reviewer 445Q and Reviewer JNfg. Compared with other generative models such as GANs, GMM costs negligible communication cost and does not introduce additional training cost.
>
> **A4**. [*A more rigorous analysis can be provided to formalize Section 5.3 findings.*]
> - Thank you for this very valuable suggestion! Intuitively, the separability of Gaussian mixtures reflects the separability of learned representations. As we know, the separability of learned representations is positively correlated with the accuracy upper bound of classifier calibration. So, the separability of Gaussian mixtures may be also positively correlated with the accuracy upper bound of classifier calibration. To quantify the separability of GMM, we can resort to [6] to calculate the Overlap Rate (OLR) of GMM. In the revision, we will provide a rigorous analysis of the relationship between the OLR of GMM and the accuracy upper bound of classifier calibration.
>
> **A5**. [*The novelty is not very high.*]
> - We would like to clarify that our work is novel and non-trivial.
>    - Existing works in FL on non-IID data lack a deep understanding of how the data heterogeneity affects each layer of a deep classification model. Our work bridges this gap by performing **the first systematic empirical study** on the hidden representations learned by different layers.
>   - The two findings from the empirical study are insightful: (1) there exists a greater bias in the classifier than other layers, and (2) the classification performance can be significantly improved by post-calibrating the classifier after federated training. The two insights may motivate future works in the FL community, as appreciated by Reviewer B69V.
>   - Our **simple yet effective** method CCVR addresses the performance degradation problem of FL on non-IID data from **a novel perspective**: classifier debiasing, which has never been discussed in previous works. We have gone through an extensive literature review and found that existing research on FL with non-IID data has generally pursued the four directions: client drift mitigation, aggregation scheme, data sharing, and personalized federated learning. Strictly speaking, our proposed CCVR does not fall into any of the four research directions.
>   - Though the idea of using Gaussian statistics to calibrate federated learning models seems simple, our empirical results show prominent accuracy gains. As acknowledged by Reviewer 445Q, ‘**the field of FL generally lacks this type of insightful yet powerful research’**.
>   - Different from most existing works, our CCVR is a post-hoc method and does not require any modification to the existing FL training procedure. Our method is general in the sense that **it can be integrated with most FL learning procedures to get a free lunch of accuracy gains**.
>
>
> [1] Group Knowledge Transfer: Federated Learning of Large CNNs at the Edge @NeurIPS 2020
>
> [2] Deep Image Prior @CVPR 2018
>
> [3] Advances and Open Problems in Federated Learning @Foundations and Trends in Machine Learning 2021
>
> [4] Federated visual classification with real-world data distribution @ECCV 2020
>
> [5] Federated learning with personalization layers @Arxiv 2019
>
> [6] Measuring the component overlapping in the Gaussian mixture model @Data Mining and Knowledge Discovery 2011

---

> > ### Comment · Reviewer_BVNc · 2021-08-23
> > **Remaining outstanding issues**
> >
> > Thanks for the detailed response to the comments. However, there are still some outstanding issues that I do not think are well addressed. Therefore, my score remains.
> >
> > > Transmitting hidden representations from actual samples is common in existing FL literature, such as [1]. However, each raw representation corresponds to a single input sample, so it may easily leak information about the client’s single examples. We hope CCVR can protect privacy at the basic level by only uploading the local Gaussian statistics instead of the raw representation. To verify this, we experiment with a popular model attack method [2] to check whether the raw image can be reconstructed by inverting the representation through the pre-trained model parameters. The visual results and detailed analysis will be included in the revision. We now share our initial findings: 1) The image recovered from the raw representation is similar to the corresponding raw image. One can generally identify the type of object. 2) The image recovered from the virtual representation generated by CCVR or gaussian mean looks largely different from the user’s raw images. It’s hard to recognize the objects in the reconstructed images. To conclude, our experiments show that transmitting local distribution statistics is more privacy-preserving than transmitting raw representations when facing attack methods like [2].
> >
> > My concern is not that GMM is better than [1] at preserving privacy, because raw representations do not protect privacy at all. Of course, local distribution statistics are more privacy-preserving than raw representations. However, leaking sufficient statistics about local populations is hardly privacy-preserving at all, especially if the data is truly (locally) Gaussian.
> >
> > > We choose GMM mainly because it is simple and effective. This merit is also appreciated by Reviewer 445Q and Reviewer JNfg. Compared with other generative models such as GANs, GMM costs negligible communication cost and does not introduce additional training cost.
> >
> > "Effective" is only true if the local data distributions are approximately Gaussian. I can hardly imagine that this is the case with FL for language models where the local linguistic contexts are complex and deviate significantly from the Gaussian. As for costs, the proposed GMM method still uses the _hidden representations_ of deep models. One has to still tune deep representations with new data, which can be just as costly as training GAN representations. To be more precise, if GANs can also be viewed as learning hidden representations to satisfy a linearly separable binary classification problem at the end (as opposed to fitting a GMM on the last layer). Additionally, the cost of fitting the GMM also scales quadratically with an increase in the number of mixtures (i.e., the number of individual nodes in the network).
> >
> > > Thank you for this very valuable suggestion! Intuitively, the separability of Gaussian mixtures reflects the separability of learned representations. As we know, the separability of learned representations is positively correlated with the accuracy upper bound of classifier calibration. So, the separability of Gaussian mixtures may be also positively correlated with the accuracy upper bound of classifier calibration. To quantify the separability of GMM, we can resort to [6] to calculate the Overlap Rate (OLR) of GMM. In the revision, we will provide a rigorous analysis of the relationship between the OLR of GMM and the accuracy upper bound of classifier calibration.
> >
> > We look forward to the said revisions.
> >
> > > We would like to clarify that our work is novel and non-trivial.
> >
> > > * Existing works in FL on non-IID data lack a deep understanding of how the data heterogeneity affects each layer of a deep classification model. Our work bridges this gap by performing the first systematic empirical study on the hidden representations learned by different layers.
> > >
> > >  * The two findings from the empirical study are insightful: (1) there exists a greater bias in the classifier than other layers, and (2) the classification performance can be significantly improved by post-calibrating the classifier after federated training. The two insights may motivate future works in the FL community, as appreciated by Reviewer B69V.
> > >
> > > * Our simple yet effective method CCVR addresses the performance degradation problem of FL on non-IID data from a novel perspective : classifier debiasing, which has never been discussed in previous works. We have gone through an extensive literature review and found that existing research on FL with non-IID data has generally pursued the four directions: client drift mitigation, aggregation scheme, data sharing, and personalized federated learning. Strictly speaking, our proposed CCVR does not fall into any of the four research directions.
> > >
> > > * Though the idea of using Gaussian statistics to calibrate federated learning models seems simple, our empirical results show prominent accuracy gains. As acknowledged by Reviewer 445Q, ‘the field of FL generally lacks this type of insightful yet powerful research’ .
> > >
> > > * Different from most existing works, our CCVR is a post-hoc method and does not require any modification to the existing FL training procedure. Our method is general in the sense that it can be integrated with most FL learning procedures to get a free lunch of accuracy gains.
> >
> > I would disagree that the conclusions drawn from this "systemic empirical study" are insightful. For example, we really do not know how well the GMM + calibration method works if the local (nodal) distributions of the hidden representations are Gaussian. If you used a different architecture (e.g., temporal architectures like LSTM or graph/transformer type representations), how often do we get locally Gaussian hidden representations in the last layers? How rapidly do we expect performance to degrade as we deviate more from local Gaussianity? Sure, the classifier debiasing emphasis can be "simple but effective", but the assumptions behind this are very strong and not very well stated in this work.

---

> > > ### Author Response · Authors · 2021-08-24
> > > **Part 1: Reply to the “Remaining outstanding issues” and new results**
> > >
> > > Thanks very much for your further reply and clarification on your concerns. We are glad to see our initial rebuttal has addressed your concern about the experimental baselines. **It seems that the “remaining outstanding issues” you pointed out are all related to the Gaussian assumption we made in the "Virtual Representations Generation" step in our method CCVR**.
> > >
> > > First, we would like to emphasize that Gaussin Distribution is only an assumption we adopted when modeling the real feature distribution. We just followed the existing literature [1][2][3] which has shown this common assumption works well. It is not the key emphasis of our work. We acknowledge that the Gaussian assumption may not be perfect to model the real feature distribution, and there must be other alternative assumptions worth exploring in the future.
> > >
> > > Second, **we would like to highlight our key contributions, which are far more significant than just a Gaussian assumption** of local feature distribution. 1) First, we provide a novel perspective to understand the non-IID quagmire in federated learning (FL). We are the first to verify the existence of classifier bias in FL through systematic experimental studies. We also show that the accuracy reduction of FL classification model can be largely alleviated by a simple classifier post-calibration technique with raw representations. 2) Second, to solve the possible privacy issues caused by transmitting the raw representations to the server, we propose a simple but effective classifier calibration algorithm CCVR. CCVR is a post-hoc method and can be combined with most FL learning algorithms to get a free lunch of considerable accuracy gains. We are glad that the other three reviewers all acknowledged the novelty and the significance of our work. Specifically, the reviewer B69V commented that our insight "is very interesting and could motivate future work in this area", the reviewer 445Q commented that "The field of FL generally lacks this type of insightful yet powerful research", and the reviewer JNfg commented that our work "presents a novel research direction". As an initial point, we believe our work could inspire interesting future works in the FL community.
> > >
> > > We will address your remaining concerns below.
> > >
> > > **A1**. [*My concern is not that GMM is better than [1] at preserving privacy, because raw representations do not protect privacy at all. Of course, local distribution statistics are more privacy-preserving than raw representations. However, leaking sufficient statistics about local populations is hardly privacy-preserving at all, especially if the data is truly (locally) Gaussian.*]
> > > - First, we should note that the reviewer’s argument about information leakage "leaking sufficient statistics about local populations is hardly privacy-preserving at all, especially if the data is truly (locally) Gaussian” has two premises. (1) One is that "data is really Gaussian distribution". As we know, it is extremely difficult to hold in real applications, especially when the images are processed by a complex deep neural network. Our extensive experiments further validate that this is not true. As shown by Table 2 and Table 3 in our submission, GMM cannot reach the accuracy upper bound achieved by using raw representations. This suggests that the local data distribution is not Gaussian at all. (2)  Another premise is “sufficient statistics”. If the first premise does not even hold, that is, the data distribution is not Guassian, using only feature means and covariances to approximate the raw representation obviously does not leak “sufficient” statistics. As far as we know, recent FL research works on privacy attacks [4][5] have focused on reconstruction attack, that is, on whether the raw local images can be recovered. We just learned that visual results such as images can be provided through external links during rebuttal. We now supplement the visual results. In the following link https://drive.google.com/file/d/1mUdq0AxTU8SXp0BwFuzEriHH-EmaaLiH/view?usp=sharing,
> > > We experimented with a popular model attack method [6] to check whether the raw images can be reconstructed by inverting the representations through the pre-trained model parameters. As we can see, the images recovered from local Guassian means are very different from the original images, thus ensuring that the raw images will not be leaked.
> > >
> > > - Second, we are glad to see that reviewer recognizes that GMM is better than existing work [7] at privacy-preserving aspect and transmitting local distribution statistics is more privacy-preserving than transmitting raw representations. That is the very merit we pursue when we design our method CCVR. The focus of our work is not on privacy, so we only claimed that our CCVR protects privacy at a basic level by uploading only the local Gaussian statistics instead of the raw representation. Note that our CCVR is just a post-hoc method, so it can be easily combined with some privacy protection techniques [8] to further secure privacy.
> > > - **To conclude, our method CCVR does not leak "sufficient" statistics about local features at all. By using CCVR, we can achieve considerable accuracy gains and the original images are difficult to be reconstructed by common attack method like [6].**
> > >
> > >
> > > **A2**. [*As for costs, the proposed GMM method still uses the hidden representations of deep models. One has to still tune deep representations with new data, which can be just as costly as training GAN representations. To be more precise, if GANs can also be viewed as learning hidden representations to satisfy a linearly separable binary classification problem at the end (as opposed to fitting a GMM on the last layer). Additionally, the cost of fitting the GMM also scales quadratically with an increase in the number of mixtures (i.e., the number of individual nodes in the network).*]
> > > - We would like to point out that the reviewer’s argument that “the proposed GMM method … can be just as costly as training GAN representations.” is problematic. We have compared the cost of training a simple Conditional GAN (CGAN) and fitting a GMM on the same features. The architecture of the CGAN follows [9]. The evaluation metrics are the number of FLOPs and the number of parameters. We randomly select a client with 3797 samples of 6 classes from CIFAR-10. To make a fair comparison, we train the conditional GAN for just 1 epoch on the local data. The dimensionality of the latent space of conditional GAN is 100. From the results below, we can see that the training cost of GAN is much higher than that of GMM. Note that we only consider local training on one client. However, the training of GANs could be more expensive in terms of communication and computation, as it usually requires collaborative training among clients and costs several communication rounds [10]. By contrast, as shown in Algorithm 1 in our submission, the procedure of fitting a GMM can be accomplished in just one communication round.
> > >
> > > |Method| Computation Cost (FLOPs)|	Number of Parameters|
> > > |  ----  | :----: | :----: |
> > > |GMM     |	0.37G |	0.97M  |
> > > |GANs    |	**7.87G** |	**1.04M**  |
> > >
> > > - Moreover, we would like to point out that the argument "the cost of fitting the GMM also scales quadratically with an increase in the number of mixtures (i.e., the number of individual nodes in the network) " may be problematic. We are not sure if by “the number of mixtures” the reviewer means "the number of classes C" in Algorithm 1 of our submission. If so, according to Equation 2, a more reasonable argument would be "the cost of fitting a GMM scales linearly with an increase in the number of mixtures and quadratically with an increase in the dimensionality of the feature". Fortunately, the dimensionality of feature $d$ is a hyperparameter that can be tuned to reduce the computation cost.
> > >
> > > - **To conclude, the adopted GMM has an advantage over GANs in terms of the computation and communication cost.**
> > >
> > >
> > > [1] Finite mixture models @ Annual review of statistics and its application 2019
> > >
> > > [2] Mixture models: theory, geometry and applications @NSF-CBMS regional conference series in probability and statistics 1995
> > >
> > > [3] One-Shot Learning with a Hierarchical Nonparametric Bayesian Model @ ICML workshop 2012
> > >
> > > [4] How to backdoor federated learning. @AISTATS 2020
> > >
> > > [5] Inverting Gradients - How easy is it to break privacy in federated learning? @NeurIPS 2020
> > >
> > > [6] Deep Image Prior @CVPR 2018
> > >
> > > [7] Group Knowledge Transfer: Federated Learning of Large CNNs at the Edge @NeurIPS 2020
> > >
> > > [8] No Peek: A Survey of private distributed deep learning @ Arxiv 2018
> > >
> > > [9] https://github.com/eriklindernoren/PyTorch-GAN
> > >
> > > [10] Data-Free Knowledge Distillation for Heterogeneous Federated Learning @ ICML 2021
> > >
> > > [11] Measuring the component overlapping in the Gaussian mixture model @Data Mining and Knowledge Discovery 2011
> > >
> > > [12] Generalized Sliced Wasserstein Distances @ NeurIPS 2019

---

> > > ### Author Response · Authors · 2021-08-24
> > > **Part 2: Reply to the “Remaining outstanding issues” and new results**
> > >
> > > **A3.** [*"Effective" is only true if the local data distributions are approximately Gaussian. I can hardly imagine that this is the case with FL for language models where the local linguistic contexts are complex and deviate significantly from the Gaussian.*]
> > >
> > > - As for the Gaussian assumption of the feature distribution, we just follow the existing literature [1][2][3]. As stated in Section 3.1 “Problem Setup”, we **follow the common setting considered by most of the related works and our considered baselines,** focusing on the image classification task in federated learning. Our extensive experiments validate that the Gaussian assumption works well for the visual model.
> > > - As for the language model, we all know that the language tasks and visual tasks are largely different. “How to approximate the language features?” is a general and open question in deep learning and is hard to be addressed in just one conference paper. We believe the question itself and the extension of our CCVR to language models are very interesting future works to explore.
> > >
> > > **A4.** [*I would disagree that the conclusions drawn from this "systemic empirical study" are insightful. For example, we really do not know how well the GMM + calibration method works if the local (nodal) distributions of the hidden representations are Gaussian. If you used a different architecture (e.g., temporal architectures like LSTM or graph/transformer type representations), how often do we get locally Gaussian hidden representations in the last layers? How rapidly do we expect performance to degrade as we deviate more from local Gaussianity?*]
> > > - The reviewer argues that the conclusions drawn from our "systemic empirical study" (Section 3) are not insightful, but the provided examples are all about the Gaussian assumption we made in the Virtual Representations Generation step in our method CCVR (Section 4). We believe that the reviewer still doesn’t catch the core insights and contributions of our work. **One should note that the contributions of the two sections are orthogonal**. As we stated in the original rebuttal, the “systematic empirical study” in Section 3 mainly focuses on “the hidden representations learned by different layers”. This empirical study gives us two important conclusions: (1) There exists a greater bias in the classifier than other layers (Section 3.2). (2) The performance degradation can be significantly alleviated by post-calibrating the classifier after federated training using IID data (Section 3.3). **We are pretty sure that these two findings are insightful since they motivate us to consider how to simulate feature distribution and apply the features to classifier calibration. Moreover, the two insights have never been discussed in previous FL literature** and may motivate future works in the FL community, as appreciated by all other reviewers.
> > >
> > > - We now address the questions raised by the reviewer one by one.
> > >   - *“We really do not know how well the GMM + calibration method works if the local (nodal) distributions of the hidden representations are Gaussian.”*: As stated in the first point of A1, the argument that “the local (nodal) distributions of the hidden representations are Gaussian.” does not hold in the real-world application, so it should no longer be a concern. **One can only claim that the feature distribution could be *approximated* by GMM**.
> > >   - *“If you used a different architecture (e.g., temporal architectures like LSTM or graph/transformer type representations), how often do we get locally Gaussian hidden representations in the last layers?”*: It is a similar question to A3, please refer to the previous clarification. To the best of our knowledge, the performances of the enumerated architectures are even unclear when trained on non-IID data, since very few works in FL have investigated them. As stated before, it's hard to settle all of these questions properly and in detail in only one paper. We believe the extensions of our work to these architectures are worth exploring.
> > >   - *“How rapidly do we expect performance to degrade as we deviate more from local Gaussianity?”*: This question is related to the performance guarantee of GMM. Please refer to the second point of A5. We have provided an empirical analysis on the relationship between the separability of GMM and the accuracy upper bound of classifier calibration.
> > >
> > > **A5.** [*Sure, the classifier debiasing emphasis can be "simple but effective", but the assumptions behind this are very strong and not very well stated in this work.*]
> > > - We are glad to see the reviewer acknowledges our "classifier debiasing emphasis" is simple but effective. However, note that **the general classifier debiasing methods discussed in Section 3.3 are not based on the Gaussian assumption.** We aim to provide a new perspective to solve the non-IID quagmire in FL. **As for the Gaussian assumption in our preliminary solution CCVR, we would like to re-emphasize that: i) It is not the key emphasis of our work. In the future, one can flexibly replace the current assumption in CCVR with a more suitable assumption to improve the performance of classifier calibration. ii) The Gaussian assumption is adopted in various existing works and we just follow them [1][2][3]. Besides, both our empirical results and the existing works show that this assumption works well.**
> > >
> > > - For **the performance guarantee of the Gaussian assumption** in the "Virtual Representations Generation" step in CCVR, we now add further analysis. We have tried to reproduce the algorithm in [11] to provide an analysis of the relationship between the Overlap Rate (OLR) of GMM and the accuracy upper bound of classifier calibration. However, the public code only supports 2-dimensional cases. So, we resort to Sliced Wasserstein Distance [12], which is a popular metric to measure the distances between distributions, to quantify the separability of GMM. The experiments are conducted on CIFAR-10 with $\alpha=0.1$. We first compute the Wasserstein distances between any two mixtures, then we average all the distances to get a mean distance. The farther the distance, the better the separability of GMM. The results are summarized below. We have also visualized the relationship between the accuracy gains and the separability of GMM through an image: https://drive.google.com/file/d/12As3XcoP1_Ml-gKr-11N8YJPz75hV6aa/view?usp=sharing. From the image and the table below, we can observe that **the mean Wasserstein distance of GMM is positively correlated with the accuracy upper bound of classifier calibration. It verifies our claim in Section 5.3: CCVR may be more effective when applied to the models with good (separable) representations. In practice, one can use the mean Wasserstein distance of GMM to evaluate the quality of the simulated representations, as well as to forecast the potential performance of classifier calibration.**
> > >
> > > |  Method   | Mean Wasserstein Distance  | Accuracy Gain of CCVR (%) |  Accuracy Gain of Whole Data (%) |
> > > |  ----  | :----: | :----: | :----: |
> > > |  FedProx |    79.00   |  3.67 | 5.70 |
> > > |  FedAvg |     82.71    | 4.13 | 6.15 |
> > > |  MOON |    97.97  | 4.86 | 7.58 |
> > >
> > > Thanks very much for your reading and patience. We sincerely hope this response and the changes we made have cleared up your remaining concerns about our work. If you have further questions about our work, please don't hesitate to post a comment. We will do our best to respond to you before the discussion period ends.
> > >
> > >
> > >
> > > [1] Finite mixture models @ Annual review of statistics and its application 2019
> > >
> > > [2] Mixture models: theory, geometry and applications @NSF-CBMS regional conference series in probability and statistics 1995
> > >
> > > [3] One-Shot Learning with a Hierarchical Nonparametric Bayesian Model @ ICML workshop 2012
> > >
> > > [4] How to backdoor federated learning. @AISTATS 2020
> > >
> > > [5] Inverting Gradients - How easy is it to break privacy in federated learning? @NeurIPS 2020
> > >
> > > [6] Deep Image Prior @CVPR 2018
> > >
> > > [7] Group Knowledge Transfer: Federated Learning of Large CNNs at the Edge @NeurIPS 2020
> > >
> > > [8] No Peek: A Survey of private distributed deep learning @ Arxiv 2018
> > >
> > > [9] https://github.com/eriklindernoren/PyTorch-GAN
> > >
> > > [10] Data-Free Knowledge Distillation for Heterogeneous Federated Learning @ ICML 2021
> > >
> > > [11] Measuring the component overlapping in the Gaussian mixture model @Data Mining and Knowledge Discovery 2011
> > >
> > > [12] Generalized Sliced Wasserstein Distances @ NeurIPS 2019

---

### Official Review · Reviewer_B69V · 2021-07-19

**Rating:** 7
**Confidence:** 3

**Summary:**

- This work proposes a classifier calibration technique to handle the data heterogeneity commonly encountered in federated learning setups.
- This work motivates its proposal by assessing the similarities of representations learned in different layers of the network, and finds that the final layer (classifier layer) is the one most dissimilar across clients.
- They propose a post-learning classifier calibration technique through representations sampled from a gaussian mixture model estimated from representations from client data.
- Experiments on CIFAR-10, CIFAR-100, and CINIC-10 with FedAvg, FedProx, and MOON show improvements over models learned with no calibration.

**Limitations And Societal Impact:**

- The authors discuss the limitations of their method in Section 5.3

**Main Review:**

**Strengths**

- The paper is very well written with a clear flow of ideas and adequate references.
- The experiments are well motivated with the prior experiment clearly leading to the next.
- The insight that the earlier layers in a network tend to learn similar features while the last layer has the most dissimilarity is very interesting and could motivate future work in this area.


**Improvements**

- Table 1: While `clsnorm` and `clsprox` are understandable on closer study, it might help to map these terms to relevant sections in Sec 3.3
- Figure 6: It will be helpful to overlay all three images on one another to enable the reader to understand the difference in heterogeneity on a common scale. Currently, with difference Y-scales for the three images, it is not easy to compare them with one another
- Figures 4 and 6: It will also be helpful to have error bars on these figures.
- There are minor typos in some places which the authors should go through and fix in the updated manuscript

**Time Spent Reviewing:**

4

---

> ### Author Response · Authors · 2021-08-10
> **Response to Reviewer B69V: Replies to the suggestions**
>
> Thanks very much for your valuable comments. We are glad that you appreciate the interesting insight, extensive experiments, and good writing of this paper. We will make the following modifications in the revised version.
> - Add detailed comments for abbreviations such as clsprox and clsnorm.
> - Improve the readability of images. We will include error bars and overlay the subfigures with the same Y-axis.
> - Go through the manuscript and fix existing typos.

---

### Decision · Program_Chairs · 2021-09-27

**Decision:**

Accept (Poster)

**Comment:**

The reviewers appreciate the simple yet effective strategy in this paper that handles heterogeneous data in federated learning. The experimental finding that the last layer exhibits the most dissimilarity could inspire future research. Great reproducibility and good writing also make the paper stand out. Therefore, I recommend acceptance. In addition, please incorporate the new results in the final version.